# E($n$)-equivariant Graph Neural Cellular Automata

**Gennaro Gala**                                                          *g.gala@tue.nl*
*Department of Mathematics and Computer Science*
*Eindhoven University of Technology, The Netherlands*

**Daniele Grattarola**                                     *daniele.grattarola@gmail.com*
*Independent researcher*

**Erik Quaeghebeur**                                                 *e.quaeghebeur@tue.nl*
*Department of Mathematics and Computer Science*
*Eindhoven University of Technology, The Netherlands*

**Reviewed on OpenReview:** *https://openreview.net/forum?id=7PNJzAxkij*

## Abstract

Cellular automata (CAs) are notable computational models exhibiting rich dynamics emerging from the local interaction of cells arranged in a regular lattice. Graph CAs (GCAs) generalise standard CAs by allowing for arbitrary graphs rather than regular lattices, similar to how Graph Neural Networks (GNNs) generalise Convolutional NNs. Recently, Graph Neural CAs (GNCAs) have been proposed as models built on top of standard GNNs that can be trained to approximate the transition rule of any arbitrary GCA. We note that existing GNCAs can violate the locality principle of CAs by leveraging global information and, furthermore, are anisotropic in the sense that their transition rules are not equivariant to isometries of the nodes' spatial locations. However, it is desirable for instances related by such transformations to be treated identically by the model. By replacing standard graph convolutions with E($n$)-equivariant ones, we avoid anisotropy by design and propose a class of isotropic automata that we call E($n$)-GNCAs. These models are lightweight, but can nevertheless handle large graphs, capture complex dynamics and exhibit emergent self-organising behaviours. We showcase the broad and successful applicability of E($n$)-GNCAs on three different tasks: (i) isotropic pattern formation, (ii) graph auto-encoding, and (iii) simulation of E($n$)-equivariant dynamical systems.

## 1 Introduction

The design of collective intelligence, i.e. the ability of a group of simple agents to collectively cooperate towards a unifying goal, is a growing area of machine learning research aimed at solving complex tasks through *emergent computation* (Ha & Tang, 2022). The interest in these techniques stems from their striking similarity to real biological systems—such as insect swarms and bacteria colonies—and from their natural scalability as distributed systems (Mitchell, 2009).

Cellular automata (CAs) (von Neumann, 1963) represent a natural playground for studying collective intelligence and morphogenesis (shape-forming processes), because of their discrete-time and Markovian dynamics (Turing, 1952). CAs are computational models inspired by the biological behaviors of cellular growth. As such, they are capable of producing complex emergent *global* dynamics from the iterative, possibly asynchronous application of *localized* transition rules (a.k.a. update rules), that can but do not need to have an analytical formulation (Adamatzky, 2010).

Research on applying neural nets for learning and designing CA rules can be traced back to Wulff & Hertz (1992), with subsequent notable contributions by Elmenreich & Fehérvári (2011), Nichele et al. (2018), and Gilpin (2019). Recently, Neural Cellular Automata (NCAs) have been proposed as CAs with transition

rules encoded as—typically light-weight—neural networks. They have been successfully applied for designing self-organizing systems for morphogenesis in 2D and 3D (Mordvintsev et al., 2020; Sudhakaran et al., 2021), image generation and classification (Palm et al., 2022; Randazzo et al., 2020), reinforcement learning (Huang et al., 2020), pathfinding and graph-diameter computation Earle et al. (2023)), and many other subdomains of machine learning. This line of work has a common theme: It assumes a fixed discrete geometry for the CA cells, which are typically arranged in $n$-dimensional, equispaced, and oriented lattices.

Subsequently, Grattarola et al. (2021) introduced GNCAs (Graph NCAs) by extending NCAs to the general setting of graphs, and showed that Graph Neural Networks are natural and universal engines for learning any desired transition rule. Their architecture, however, does not allow nodes to have hidden states, which have been proven to be useful for encoding perception and evolution history (Mordvintsev et al., 2020). More crucially, their formulation allows nodes to be *aware of their global locations* and sticks to a *fixed frame of reference*, therefore ignoring the possible symmetries in the state space even for states representing spatial information like position and velocity.

By building on E($n$)-equivariant Graph Neural Networks (Satorras et al., 2021b), we elegantly overcome these relevant issues and present GNCAs that respect isometries in the state space *by design*, leading to truly self-organizing systems. Our contributions are twofold:

- We propose the first isotropic-by-design GNCAs, which we name E($n$)-GNCAs. More specifically, these models are E($n$)-equivariant and, crucially, **cannot** globally localise the nodes. In this way, unlike standard GNCAs, it is impossible to violate the core locality principle of CAs, making it **essential** to solve a unifying, shared goal;

- We provide extensive guidelines on how to train E($n$)-GNCAs and showcase their broad, successful applicability on three different tasks: (i) pattern formation, (ii) graph auto-encoding, and (iii) simulation of (self-organizing) E($n$)-equivariant dynamical systems.

Our model and results represent a step forward in the design of self-organizing neural systems and can have concrete impact in modeling and understanding natural systems governed by strong local interactions, ranging from chemical to social phenomena (Ha & Tang, 2022).

## 2 Preliminaries and Related Work

We here introduce necessary concepts of and relevant prior work on cellular automata and (equivariant) graph neural networks. These support and contextualize the model we propose.

**Graphs** A graph $\mathcal{G} = (\mathcal{V}, \mathcal{E})$ consists of an unordered set of nodes $\mathcal{V} = \{1, \dots, |\mathcal{V}|\}$ and a set of edges $\mathcal{E} \subseteq \mathcal{V} \times \mathcal{V}$. Its neighbourhood function $\mathcal{N}$ is defined for every node $i \in \mathcal{V}$ by $\mathcal{N}(i) = \{j \in \mathcal{V} : (i,j) \in \mathcal{E}\}$. A graph can be equivalently defined with an adjacency matrix $A \in \{0,1\}^{|\mathcal{V}| \times |\mathcal{V}|}$, where $A_{ij}$ is 1 if and only if $(i,j) \in \mathcal{E}$.

We can attach a state $\mathbf{s}_i \in \mathcal{S}$ to each node $i$ and an attribute $\mathbf{e}_{ij} \in \mathcal{A}$ to each edge $(i,j)$, where for now we leave the state space $\mathcal{S}$ and attribute set $\mathcal{A}$ unspecified. A node state $\mathbf{s}_i$ typically consists of components such as location $\mathbf{x}_i$, velocity $\mathbf{v}_i$, and (hidden) node features $\mathbf{h}_i$. Jointly for all nodes and edges, we write $\mathbf{S}$—with components $\mathbf{X}$, $\mathbf{V}$, and $\mathbf{H}$—and $\mathbf{E}$ respectively, which implicitly carry with them the underlying graph.

### 2.1 Graph (Neural) Cellular Automata

**Graph Cellular Automata** A *Graph Cellular Automaton* (GCA) is a triple $(\mathcal{G}, \mathcal{S}, \tau)$, where $\mathcal{G} = (\mathcal{V}, \mathcal{E})$ is a graph and $\mathcal{S}$ is a discrete or continuous state space. The map $\tau : \mathcal{S} \times 2^{\mathcal{S}} \to \mathcal{S}$ is used as a *local* transition rule to update the state $\mathbf{s}_i \in \mathcal{S}$ of each of the graph's nodes $i \in \mathcal{V}$ as a function of its current state and its neighbour's states:

$$\mathbf{s}_i' = \tau\big(\mathbf{s}_i, \{\mathbf{s}_j : j \in \mathcal{N}(i)\}\big). \tag{1}$$

We will compactly write $\mathbf{S}' = \tau(\mathbf{S})$ to indicate the synchronous application of $\tau$ to all nodes in $\mathcal{G}$. Standard CAs—like elementary CAs (Wolfram, 2018) and Conway's Game of Life (Adamatzky, 2010)—use a simple

grid for the underlying graph $\mathcal{G}$, have integer-valued locations $\mathbf{x}_i \in \mathbb{Z}^n$ and use a single binary value for their features $\mathbf{h}_i$.

**Anisotropy & Isotropy**  Of great importance for CAs are the properties *anisotropy* and *isotropy*: The former implies being directionally dependent, as opposed to the latter, which indicates homogeneity in all directions. Anisotropic transition rules account for not only the neighbor states of a given node, but also their *absolute* position in a (vector) space of reference. Furthermore, anisotropic transition rules are *not* invariant to rotations, translations and reflections of the states, thus resulting in nodes being oriented in a specific direction and prohibiting the existence of differently oriented states of interest (Mordvintsev et al., 2022; Grattarola et al., 2021). In contrast, isotropy allows transition rules to act similarly regardless of how the nodes are oriented, thus allowing proper design of self-organising (and living) systems.

**Neural Cellular Automata**  A neural cellular automaton (NCA) uses a light-weight neural net with parameters $\theta$ for its transition rule $\tau_\theta$ (Mordvintsev et al., 2020). In this setting, states are represented as typically low-dimensional vectors. The differentiability of the transition rule allows for optimisation of its parameters $\theta$ via backpropagation through time (Lillicrap & Santoro, 2019). Recent work has shown the successful application of deep learning techniques for NCAs, showing that neural transition rules can be efficiently learned to exhibit complex desired behaviors (Mordvintsev et al., 2020; 2022; Tesfaldet et al., 2022; Grattarola et al., 2021; Palm et al., 2022).

Note that (N)CAs are *unaware* of time and their execution is not constrained by a finite time interval. Furthermore, they can only be inspected via simulation from a state of interest and that represents a key feature of these models. For instance, elementary CAs (Wolfram, 2018), e.g. *Rule 30*, can run forever and, crucially, an arbitrary future state cannot be predicted from a current one unless via simulation, i.e. by iteratively applying the transition rule up to the time step of interest.

As already pointed out by Tesfaldet et al. (2022), NCAs are not structurally equivalent to (deep) feed-forward neural nets, where an *acyclic* directed computation graph induces a *finite* impulse response. Instead, NCAs can be viewed as Recurrent Neural Networks (Rumelhart et al., 1986), where a *cyclic* directed computation graph induces an *infinite* impulse response, enabling feedback and time-delayed interactions. Notably, RNNs and CAs—and by consequence NCAs—are known to be Turing complete (Pérez et al., 2019).

## 2.2 Graph Neural Networks

Graph Neural Networks (GNNs) (Gori et al., 2005) have become the go-to method for representation learning on graphs. The core functionality of GNNs is the message-passing scheme. Let $\mathbf{s}_i \in \mathbb{R}^s$ represent the feature vector of node $i$ and $\mathbf{e}_{ij} \in \mathbb{R}^e$ the (possibly available) feature vector of edge $(i,j)$. A message-passing layer updates the features of node $i$ as follows:

$$\mathbf{s}'_i = \gamma\big(\mathbf{s}_i, \bigoplus_{j \in \mathcal{N}(i)} \phi(\mathbf{s}_i, \mathbf{s}_j, \mathbf{e}_{ji})\big), \tag{2}$$

where $\phi$ is the message function, $\bigoplus$ is a permutation-invariant operation to aggregate the set of incoming messages, and $\gamma$ is the node update function. The differentiable operators $\phi$, $\bigoplus$, and $\gamma$ allow message-passing layers to be stacked sequentially and then optimised with (stochastic) gradient descent.

## 2.3 E($n$)-equivariant Graph Neural Networks

Our work builds on E($n$)-equivariant GNNs (EGNNs) (Satorras et al., 2021b). In this setting, every graph node $i$ has coordinates $\mathbf{x}_i \in \mathbb{R}^n$ and node features $\mathbf{h}_i \in \mathbb{R}^h$, and an edge $(i,j) \in \mathcal{E}$ can possibly have attributes $\mathbf{e}_{ij} \in \mathbb{R}^e$. EGNNs represent a class of GNNs explicitly designed to be permutation equivariant with respect to the nodes (like any GNN), and translation, rotation and reflection equivariant with respect to nodes' coordinates. The isometry group corresponding to these symmetries is called the Euclidean group E($n$). We will formally discuss the *key features* of EGNNs while presenting our method in Section 3.

**E($n$)-equivariant Graph Convolutions**  Given a graph $\mathcal{G}$, node coordinates $\{\mathbf{x}_i\}$, node features $\{\mathbf{h}_i\}$ and *optional* edge attributes $\{\mathbf{e}_{ij}\}$ an E($n$)-equivariant Graph Convolution (EGC) sequentially performs:

$$\mathbf{m}_{ij} = \phi_m(\|\mathbf{x}_i - \mathbf{x}_j\|^2, \mathbf{h}_i, \mathbf{h}_j, \mathbf{e}_{ij}) \quad (3) \qquad \mathbf{x}_i' = \mathbf{x}_i + \frac{1}{|\mathcal{N}(i)|} \sum_{j \in \mathcal{N}(i)} (\mathbf{x}_i - \mathbf{x}_j)\phi_x(\mathbf{m}_{ij}) \quad (5)$$

$$\mathbf{m}_i = \sum_{j \in \mathcal{N}(i)} \mathbf{m}_{ij} \quad (4) \qquad\qquad \mathbf{h}_i' = \phi_h(\mathbf{h}_i, \mathbf{m}_i) \quad (6)$$

where $\phi_m : \mathbb{R}^{1+2h+e} \to \mathbb{R}^m, \phi_x : \mathbb{R}^m \to \mathbb{R}^1$ and $\phi_h : \mathbb{R}^{h+m} \to \mathbb{R}^{h'}$ are MLPs. Concisely, we write $\mathbf{X}', \mathbf{H}' = \text{EGC}(\mathbf{X}, \mathbf{H}, \mathbf{E})$. If $h' = h$, a skip connection can be used in Equation 6 as follows:

$$\mathbf{h}_i' = \phi_h^+(\mathbf{h}_i, \mathbf{m}_i) = \phi_h(\mathbf{h}_i, \mathbf{m}_i) + \mathbf{h}_i. \quad (7)$$

**E($n$)-equivariant Graph Convolutions with Attention**   To assign different weights when aggregating messages, we can use attention and replace Eq. 4 with:

$$\mathbf{m}_i = \sum_{j \in \mathcal{N}(i)} \phi_a(\mathbf{m}_{ij})\mathbf{m}_{ij} \quad (8)$$

where $\phi_a : \mathbb{R}^m \to [0,1]^1$ is an MLP that takes a message $\mathbf{m}_{ij}$ as input and outputs its attention weight $\phi_a(\mathbf{m}_{ij})$. We will use this formulation in Section 4.3. Note that, attention weights are particularly advantageous when a very dense (possibly fully connected) graph is used (Satorras et al., 2021b; Vaswani et al., 2017).

**E($n$)-equivariant Graph Convolutions with Velocity**   When nodes represent bodies with velocities, we can extend the previous formulation to explicitly account for velocity. Given node velocities $\{\mathbf{v}_i\}$ we can replace the coordinate update in Equation 5 with the following two steps:

$$\mathbf{v}_i' = \phi_v(\mathbf{h}_i, \|\mathbf{v}_i\|)\mathbf{v}_i + \frac{1}{|\mathcal{N}(i)|} \sum_{j \in \mathcal{N}(i)} (\mathbf{x}_i - \mathbf{x}_j)\phi_x(\mathbf{m}_{ij}) \quad (9) \qquad \mathbf{x}_i' = \mathbf{x}_i + \mathbf{v}_i' \quad (10)$$

where $\phi_v : \mathbb{R}^{h+1} \to \mathbb{R}^1$ is an MLP. Without affecting equivariance, and different from Satorras et al. (2021b), $\phi_v$ has not only $\mathbf{h}_i$, but also $\|\mathbf{v}_i\|$ as an argument, since we found it to be very beneficial in practice (cf. Section 4.3). Concisely, we write $\mathbf{X}', \mathbf{V}', \mathbf{H}' = \text{EGC}(\mathbf{X}, \mathbf{V}, \mathbf{H}, \mathbf{E})$.

**E($n$)-equivariant GNNs**   An E($n$)-equivariant GNN is a stack of $\ell \geq 1$ EGCs applied sequentially. Concisely, we write $\mathbf{X}', \mathbf{H}' = \text{EGNN}_\ell(\mathbf{X}, \mathbf{H}, \mathbf{E})$ to denote the application of an EGNN with $\ell$ layers.

## 3   E($n$)-equivariant Graph Neural CAs

Our work builds on the connection between isotropic GCAs and EGNNs. This becomes apparent by comparing Equation 1 with Equations 5 and 6. Specifically, we consider a setting in which a *neural parametrised* transition rule $\tau_\theta$ is implemented with a single E($n$)-equivariant Graph Convolution (EGC) acting on a continuous state space $\mathcal{S} \equiv \mathbb{R}^{n+h}$, or $\mathcal{S} \equiv \mathbb{R}^{2n+h}$ when velocity is included. We call such model E($n$)-equivariant GNCAs, E($n$)-GNCAs for short. Similarly to standard CAs, $\tau_\theta$ is repeatedly applied over time:

$$\mathbf{X}', \mathbf{H}' = \tau_\theta^t([\mathbf{X}, \mathbf{H}]) = \underbrace{\tau_\theta \circ \cdots \circ \tau_\theta}_{t \text{ times}}([\mathbf{X}, \mathbf{H}]), \quad (11)$$

where $\mathbf{X}$ represents input node coordinates and $\mathbf{H}$ represents input node features. Note that (i) to avoid clutter in Equation 11 we did not consider possibly available edge attributes $\mathbf{E}$, (ii) a similar formulation is possible when velocities $\mathbf{V}$ are available (cf. Equation 9), and (iii) the dependency of $\tau_\theta$ on a static graph $\mathcal{G}$ is left implicit in order to keep notation uncluttered. The overall state configuration $\mathbf{S}$ of an E($n$)-GNCA is defined as $\mathbf{S} = [\mathbf{X}, \mathbf{H}]$, or $\mathbf{S} = [\mathbf{X}, \mathbf{H}, \mathbf{V}]$ when velocity is available, and consequently we denote the $t$-times application of the model transition rule as $\mathbf{S}' = \tau_\theta^t(\mathbf{S})$. Note that the transition rules we consider is 1-step Markovian, meaning that automaton state at step $t+1$ is fully determined by the state at step $t$.

**On the single-layered architecture**   Using a single layer for $\tau_\theta$ entails (i) optimizing less parameters and (ii) having the strictest possible locality bottleneck, i.e. nodes are *exclusively* influenced by their immediate surroundings and do not have direct access to the global state of the entire system. In fact, using more layers would make the tasks we will study (cf. Section 4) less local—and *less* challenging—because the model

can account for a larger and more informative receptive field when transitioning from $\mathbf{S}_t$ to $\mathbf{S}_{t+1}$. This is a common design choice in NCA literature (Mordvintsev et al., 2020; 2022; Palm et al., 2022; Grattarola et al., 2021). However, a layered EGNN is still a viable approach for $\tau_\theta$ if we want to account for higher-order neighbours when performing a state update, as in the artificial life system Lenia (Chan, 2019).

**E($n$)-equivariance, E($n$)-invariance and Isotropy**
Analogously to plain EGNNs (Satorras et al., 2021b), for any positive integer $t \in \mathbb{N}^+$, orthogonal matrix $Q \in \mathbb{R}^{n \times n}$ and translation vector $b \in \mathbb{R}^n$, our neural transition rule $\tau_\theta$ satisfies

$$\psi(\mathbf{X}'), \mathbf{H}' = \tau_\theta^t([\psi(\mathbf{X}), \mathbf{H}]), \qquad (12)$$

where $\mathbf{X}', \mathbf{H}' = \tau_\theta^t([\mathbf{X}, \mathbf{H}])$ and $\psi(\mathbf{X}) = Q\mathbf{X} + b$ is short-hand for $(Q\mathbf{x}_1 + b, \dots, Q\mathbf{x}_{|\mathcal{V}|} + b)$ [1]. The function $\psi$ is an *isometry*, and represents a rotation-reflection-translation of the coordinates. As such, $\psi$ preserves the Euclidean distance between every pair of nodes. In other words, applying $\psi$ to input coordinates $\mathbf{X}$ and then running transition rule $\tau_\theta^t$ will give the same results as first running $\tau_\theta^t$ and then applying $\psi$ to $\mathbf{X}'$, as shown in Figure 1. As a consequence, output coordinates $\mathbf{X}'$ and output node features $\mathbf{H}'$ are respectively E($n$)-equivariant and E($n$)-invariant to isometries of input coordinates $\mathbf{X}$. Intuitively, these properties are a consequence of only processing relative positions and never being aware of absolute node locations (cf. Equations 3 and 5).[2][3] The E($n$)-invariance of the node features and the E($n$)-equivariance of the node coordinates make E($n$)-GNCAs isotropic *by design*.

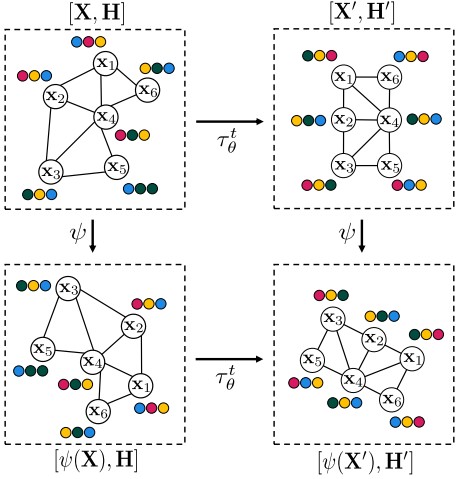

Figure 1: E($n$)-GNCA commutative diagram: For any number of steps $t$ the transition rule $\tau_\theta$ is run, output coordinates $\mathbf{X}'$ and node features $\mathbf{H}'$ are respectively E($n$)-equivariant and E($n$)-invariant to isometries of input coordinates $\mathbf{X}$. Node features are represented with 3 colored dots.

**Hidden States & Perception** Similarly to Mordvintsev et al. (2020; 2022), Palm et al. (2022), and Chan (2019), but *different* from Grattarola et al. (2021), our model has the necessary inductive bias for modelling hidden states, as it offers location-independent node features $\mathbf{H}$. These features are crucial because they can encode past evolutionary history as well as higher-order geometric information. This is not possible with original GNCAs, where node locations represent the whole state of the system (i.e. $\mathbf{X} = \mathbf{S}$), without hidden states allowing to encode other kind of information. As Mordvintsev et al. (2020), we interpret our hidden states as a signal mechanism for orchestrating morphogenesis: All nodes share the same genome, i.e. the transition rule, and only differ from the information encoded by the signaling they receive, emit, and store internally, i.e. their node features. In case node features $\mathbf{H}$ are *not* available in advance, we can either set them to $\mathbf{1}$ or randomly initialize them, and give the model the freedom to learn and use them while evolving. Further, messages $\{\mathbf{m}_{ij}\}$ (cf. Equation 3) are similar in spirit to the perception vectors of Mordvintsev et al. (2020; 2022), as they encode what nodes perceive of the environment from communicating with their neighbors.

Given (i) the interest in what would happen as $t \to \infty$ and (ii) the recurrent architecture of our model, we normalise node feature $\mathbf{H}$ after each transition rule application so as to mitigate problems like over-smoothing, exploding/vanishing gradients, and training instabilities. Specifically, after every transition rule application, we normalise node features $\mathbf{H}$ with either PairNorm (Zhao & Akoglu, 2020) or NodeNorm (Zhou et al., 2021), helpful *parameter-free* normalisation techniques for deep GNNs. Furthermore, we use the hyperbolic tangent TanH() as non-linear activation function—a common design choice in RNNs (Lipton et al., 2015)—and the skip connection defined in Equation 7, which has proven to be beneficial for deep GNNs (Xu et al., 2021; Zhao & Akoglu, 2020).

---

[1] Note that if velocity $\mathbf{V}$ is involved, it would be transformed as $Q\mathbf{V}$.

[2] Satorras et al. (2021b, Appendix) provide a formal proof of the equivariance/invariance of EGNNs.

[3] In the image domain, similar behavior is exhibited by CNN-based NCAs: Rotating the perceptive field of the Sobel convolutional kernels leads to equivalently rotated target images (Mordvintsev et al., 2020, experiment 4).

**Global propagation from local interactions** Message-passing GNNs require $\ell$ layers to allow communication between nodes that are $\ell$-hops away. Several tasks in graph ML tend to be very challenging when the diameter of the underlying graph $\mathcal{G}$ is larger than the number of layers used, and that is because the receptive field of the network may not comprise the whole graph (Zhao & Akoglu, 2020; Alon & Yahav, 2021). Further, to avoid severe over-smoothing (Li et al., 2018), most popular GCN-style networks (Kipf & Welling, 2017) tend to be shallow, with narrow receptive fields, leading to *under-reaching* (Wenkel et al., 2022). To avoid this limitation, it is common to exchange messages among all nodes and provide the edge information $(i, j) \in \mathcal{E}$ as a Boolean flag within the edge attributes (Liu et al., 2019; Satorras et al., 2021b;a). This is computationally quadratic in the number of nodes, and therefore very challenging and computationally expensive when processing large graphs.

In our setting, due to the 1-step Markovian property of $\tau_\theta$, the effective receptive field of E($n$)-GNCA localized message-passing grows larger with each state update until eventually encompassing the whole graph. In this way global propagation of information arises from localized interactions of nodes. In other words, iterative local message-passing circumvents the quadratic complexity and related challenges of exchanging messages among all nodes at each step. This self-organizing process does not require any external control or centralized leader: nodes communicate with their neighbors to make collective decisions about the final configuration of the nodes. This globally consistent and complex behaviour, which arises from strictly local interactions, is a particular feature of (N)CAs as we show in our experiments. Finally, we emphasise that—despite the localized model computation—we are allowed to express global information within the loss function used for training. In other words, the training signal can account for the distance between two nodes that are actually not connected via an edge, as we will do in Equations 13 and 15.

**On locality & time** Very often in the literature, GNN computation graphs are *decoupled* from the input graphs. For instance, Satorras et al. (2021b) and Satorras et al. (2021a) use *layered* EGNNs with fully-connected computation graphs despite input graphs being sparse (e.g. molecules). E($n$)-GNCAs, instead, always use *sparse computation graphs*. Moreover, GNN-based diffusion models—and often also neural simulator (Chen et al., 2018)—are *aware* of time, which is usually included in node features via concatenation (Hoogeboom et al., 2022) and that allows training using mini-batches of sub-trajectories extracted at different time steps (Yang et al., 2022). E($n$)-GNCAs are instead *unaware* of time, making tasks only solvable through simulation from an initial state of interest. These are fundamental design choices which highly impact the tasks we will study. Finally, note that diffusion-based models and neural simulators often only care about a finite rollout, whereas, as we show in our experiments, we care about open-endeness, i.e. infinite rollouts.

## 4    Experiments

We showcase the successful applicability of E($n$)-GNCAs in three different tasks: (i) pattern formation, (ii) graph autoencoding and (iii) simulation of E($n$)-equivariant dynamical systems. We set $h = 16$ (hidden dimension) and $m = 32$ (message dimension) throughout *all* experiments, leading to an overall automaton size of only 5K parameters *irrespective* of the coordinate dimension being used. We are grateful to the developers of the main software packages used for this work: Pytorch (Paszke et al., 2019), PyTorch Geometric (Fey & Lenssen, 2019) and Lightning (Falcon & The PyTorch Lightning team, 2019). Our code is available at github.com/gengala/egnca. All experiments are run on an NVIDIA Quadro P1000 16GB, and each run does not take more than 2.5 hours to complete.

### 4.1    Pattern Formation

Inspired by prior work on CA morphogenesis (Mordvintsev et al., 2020; 2022; Grattarola et al., 2021), we show how E($n$)-GNCAs can be trained to converge to a given fixed target state. In our case, the target is a sparse geometric graph $\mathcal{G}$ that visually defines a recognisable 2D or 3D shape. Specifically, the goal is to learn a transition rule $\tau_\theta$ that *morphs* randomly initialised coordinates $\overline{\mathbf{X}}$ to a given target point cloud $\hat{\mathbf{X}}$ by convolving over $\mathcal{G}$ and assuming a prior 1-to-1 correspondence between nodes in $\overline{\mathbf{X}}$ and $\hat{\mathbf{X}}$.

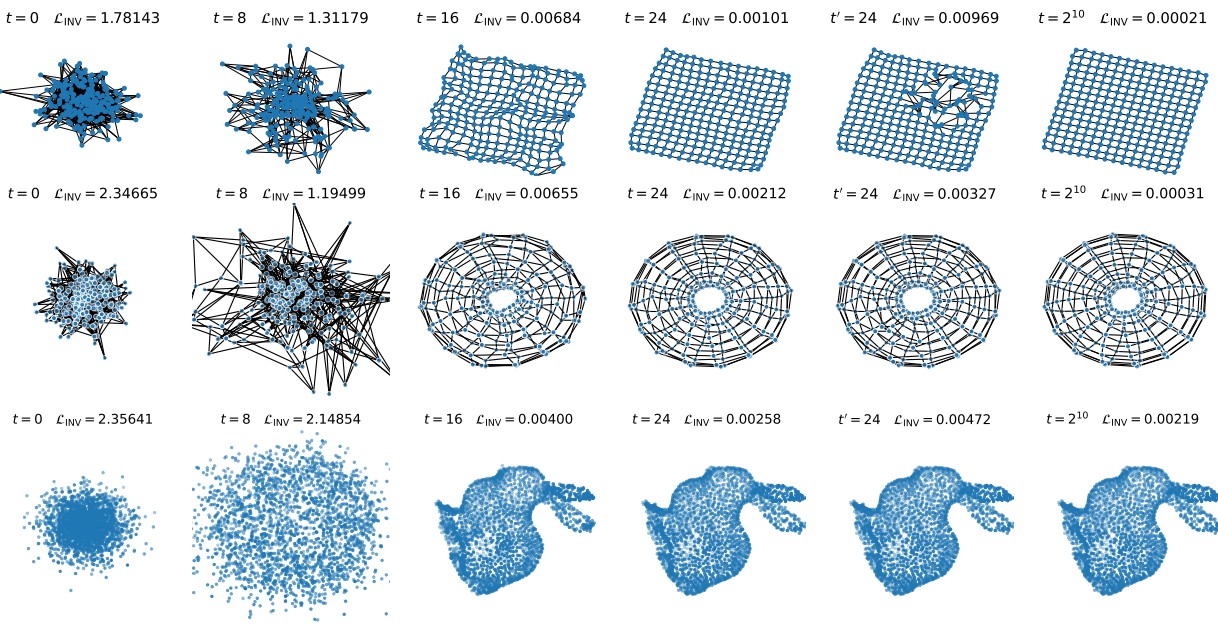

Figure 2: E($n$)-GNCA convergence to a 2D grid (top), a 3D torus (middle) and the Stanford geometric bunny (bottom). The first 4 columns show E($n$)-GNCA states at different time steps. The second to last column shows either a local or global damage of coordinates at $t = 24$. Finally, the last column shows regeneration and persistency abilities by running the transition rule for 1000 extra steps after perturbation has occurred. Note that, if we were to apply any isometry at any point in time, convergence and persistency would still be guaranteed. We report $\mathcal{L}_{\mathsf{INV}}$ (cf. Equation 13) for the state in each figure. The nearest-neighbor edges of the Stanford bunny are not shown so as to avoid clutter. We report complete trajectories in Appendix A. Best viewed digitally and zoomed in.

**E($n$)-invariant objective** Contrary to Grattarola et al. (2021), we are *not* interested in a specific orientation of $\hat{\mathbf{X}}$ and therefore we do *not* optimise the model by minimising the MSE between coordinates reached by the model and target coordinates, i.e. $\|\mathbf{X}' - \hat{\mathbf{X}}\|^2$ where $[\mathbf{X}', \mathbf{H}'] = \tau_\theta^t([\overline{\mathbf{X}}, \mathbf{1}])$. The former, moreover, would *not* be a suitable objective for our automata since it accounts for specific *global* locations which are unknown to our model that only uses relative positions during its computation (cf. Equations 3 and 5). Therefore, for every pair of nodes $(i, j) \in \mathcal{V} \times \mathcal{V}$, we minimise the MSE between their distance in the model's final configuration and the target one. Formally, we define an E($n$)-invariant objective as follows:

$$\mathcal{L}_{\mathsf{INV}} = \frac{1}{|\mathcal{V}|^2} \sum\nolimits_{(i,j) \in \mathcal{V} \times \mathcal{V}} (\|\mathbf{x}'_i - \mathbf{x}'_j\| - \|\hat{\mathbf{x}}_i - \hat{\mathbf{x}}_j\|)^2. \tag{13}$$

This objective accounts for all pairwise distances, as this is necessary to uniquely identify the target state, which cannot be done if we only considered the edges of the graph.[4] However, this does not make the model less local: Even if global information is used in the loss function, the model itself still only uses local communication to perform the task. Equation 13 provides a *much weaker* supervision signal than the one by Grattarola et al. (2021), therefore leading to a *much more* challenging task. That is because in their model every node is **aware of its global location** and has only a single constraint to satisfy, i.e. being close to a specific global location. Moreover, being aware of global location violates the locality principle of CAs and makes pattern formation rather easy to hack: We found that a 3-layer MLP with Fourier features (Tancik et al., 2020) can transform any given initial state to a target one in a few optimisation steps, without relying on a neighborhood graph. In our case, however, nodes are *not* aware of their global location, and

---

[4]For instance, consider a simple square: If we only minimize for the distances of its edges, the model could converge to a state whose loss value is zero but that does not form a square (e.g., when two opposite vertices share the same location).

have $|\mathcal{V}| - 1$ constraints to satisfy, i.e. its distances w.r.t. all the other nodes in $\mathcal{G}$, *not* only its neighbors. Interestingly, optimizing for Equation 13 results in learning a transition rule $\tau_\theta$ such that $[\mathbf{X}', \mathbf{H}'] = \tau_\theta^t([\overline{\mathbf{X}}, \mathbf{1}])$ and $\mathbf{X}' = \psi(\hat{\mathbf{X}})$ for any arbitrary isometry $\psi$. In other words, our objective gives the model the freedom to converge in any possible orientation of the target. In practice, one could avoid evaluating $\mathcal{O}(|\mathcal{V}|^2)$ distances by only considering a randomly sampled subset of node pairs when computing Equation 13. Our objective is similar in spirit to the one by Mordvintsev et al. (2022), where a rotation-reflection invariant objective is used in the image domain. Similarly, Equation 13 is an E($n$)-invariant loss function that uniquely characterizes a target point cloud up to isometries when it is equal to zero, and fits well with the model isotropy. However, the objective is *not* node-permutation invariant as it relies on a 1-to-1 correspondence between initial and target nodes: If the initial and target states were equal (up to isometries) but the correspondence between nodes did not match, the loss value would not be zero. Although this represents a limitation of the current loss function and may lead to sub-optimal dynamics, the feasibility of the task still entirely depends on local communication, which represents the core of our experimental investigation.

**Training**   We mostly follow the experimental setup in (Mordvintsev et al., 2020; Grattarola et al., 2021). First, we create a large pool (a.k.a. *cache*) of $K$ states $\{\mathbf{S}^{(k)}\}_{k=1}^K = \{[\mathbf{X}^{(k)}, \mathbf{H}^{(k)}]\}_{k=1}^K$, each initialised as $[\overline{\mathbf{X}}, \mathbf{1}]$, where $\overline{\mathbf{X}} \sim \mathcal{N}(\mathbf{0}, \sigma\mathbf{1})$. Then, we randomly sample a mini-batch from the pool and use it as input to transition rule $\tau_\theta$, which runs for a number of time steps $t$ sampled uniformly from the interval $[15, 25]^5$. Once a mini-batch is processed, we apply backpropagation through time (BPTT) (Lillicrap & Santoro, 2019) to update parameters $\theta$ according to Equation 13. To promote persistency, we use the pool as a *replay memory*, i.e. once an optimisation step is performed, we replace the pool state $\mathbf{S}^{(k)}$ with $\tau_\theta^t(\mathbf{S}^{(k)})$ for every $\mathbf{S}^{(k)}$ in the current mini-batch. This allows further training iterations to account for states that already result from a repeated application of the transition rule, thus encouraging the model to persist in the target state after reaching it. Furthermore, before processing a mini-batch, the state with the highest loss value is replaced with the initial state $[\overline{\mathbf{X}}, \mathbf{1}]$ so as to both stabilise training and, more importantly, avoid catastrophic forgetting. Finally, to also promote regeneration, we perturb half of the point clouds in the batch by adding Gaussian noise: A part is perturbed globally and another only locally.

**Results**   We consider 8 different geometric graphs, and succeed with all of them, although only a representative subset is reported in the main text due to limited space: A 2D grid (256 nodes), a 3D torus (256 nodes) and the Stanford bunny (2503 nodes) (Defferrard et al., 2017). Figure 2 shows (part of) E($n$)-GNCA trajectories as well as the loss value (Equation 13) w.r.t. the coordinates at each time step shown. Remarkably, our model learns to converge to a ***stable*** attractor of the given geometric graph, i.e. the model can maintain the target state after any number of time steps $t > 15$, as shown in the last column of Figure 2. Furthermore, the model exhibits regeneration abilities by being robust against perturbations of the nodes. More details, animations and results can be found in Appendix A and supplementary material.

**On Isotropic Pattern Formation**   Isotropy is a well-established property in CAs and it is very desirable in our context. In fact, isotropic pattern formation—besides being more challenging to achieve—is more general and subsumes its anisotropic counterpart. This means that if directional dependence is needed, we could manipulate the automata such that the target states will be placed at any desired orientation. This is not possible with original GNCAs, given their awareness of global locations, which violates locality. We refer the reader to Figure A.9 for a visual example of the concept above.

## 4.2   Graph autoencoding with Cellular Automata

In this section, we show how E($n$)-GNCAs can be deployed as performant Graph AutoEncoders (GAEs) (Kipf & Welling, 2016), despite their single-layered architecture and recurrent computation. In graph autoencoding one has available a set of (possibly featureless) graphs $\{\mathcal{G}_n\}$ and wants to learn node representations that can be used to reconstruct the underlying ground-truth adjacency matrices (Satorras et al., 2021b; Liu et al., 2019). Specifically, we will deal with graph autoencoding in Euclidean space, i.e. two nodes will be connected

---

[5]The interval considered represents a trade-off between computational complexity, stability during training, and a sufficient number of time steps to allow the model to learn dynamical patterns for the desired behavior.

if their distance is below or equal to a given threshold $\hat{t}$, which can be fine-tuned on the validation set after training. For this task, we report more details and additional results in Appendix B.

**Datasets**  We consider five datasets of featureless graphs of varying size, connectivity and properties: COMM-S (100 graphs, 2 communities, 12–20 nodes) (Liu et al., 2019), PLANAR-S (200 planar graphs, 12–20 nodes), PLANAR-L (200 planar graphs, 32–64 nodes), SBM (200 stochastic block model graphs, 2–5 communities, 44–187 nodes) (Martinkus et al., 2022) and PROTEINS (918 graphs, 100–500 nodes) (Dobson & Doig, 2003). Figure B.10 shows some examples of such graphs. Planar graphs are generated by first uniformly sampling 2D points in $[0, 1]^2$ and then applying Delaunay triangulation. We split all datasets into training (80%), validation (10%) and test (10%) sets.

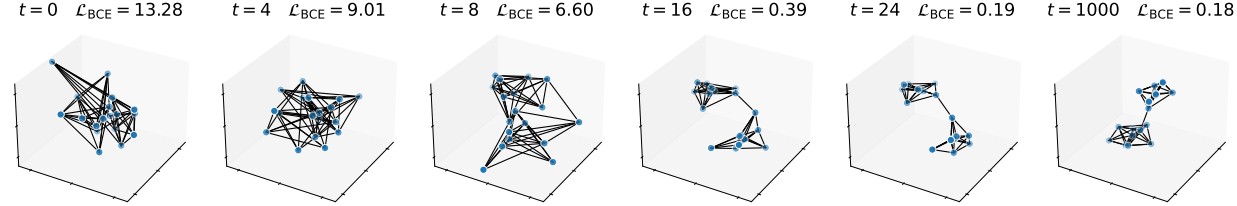

Figure 3: E($n$)-GNCA coordinates at different time steps for a test-set graph in COMM-S. In each figure, we plot the ground-truth edges and report the binary cross-entropy (cf. Equation 15).

**Training**  For each training graph $\mathcal{G}_n$ we create a small pool of $K$ states $\{[\mathbf{X}^{(n,k)}, \mathbf{H}^{(n,k)}]\}_{k=1}^{K}$. Every $\mathbf{H}^{(n,k)}$ is again initialised as $\mathbf{1}$ whereas input node coordinates $\mathbf{X}^{(n,k)}$ now follow an isotropic Gaussian $\mathcal{N}(\mathbf{0}, \sigma\mathbf{1})$.[6] As such, the model can be viewed as a generative model *conditioned* on $\mathcal{G}$. A mini-batch is now created by first considering a random subset of training graphs and then sampling a random pool state each. Every mini-batch state $\mathbf{S}^{(n,k)}$ is then run by $\tau_\theta$ for $t \in [t_1, t_2]$ random time steps, eventually reaching state $[\mathbf{X}', \mathbf{H}'] = \tau_\theta^t(\mathbf{S}^{(n,k)})$. Finally, we apply an E($n$)-invariant decoding scheme based on distances between nodes $\mathbf{X}'$ so that the reconstructed soft adjacency matrix $\hat{A} \in [0, 1]^{|\mathcal{V}| \times |\mathcal{V}|}$ is defined as:

$$\hat{A}_{ij} = \frac{1}{1 + \exp(\delta_2(\|\mathbf{x}'_i - \mathbf{x}'_j\|_2^2 - \delta_1))} \in [0, 1], \tag{14}$$

where $\delta_1$ and $\delta_2$ are learnable positive scalars. The model is trained by minimising the binary cross-entropy (BCE) between the ground-truth adjacency $A$ and the predicted soft one $\hat{A}$, namely:

$$\mathcal{L}_{\mathsf{BCE}} = -\sum_{ij} A_{ij} \ln(\hat{A}_{ij}) + (1 - A_{ij}) \ln(1 - \hat{A}_{ij}). \tag{15}$$

Equation 15 can be seen as a more relaxed version of Equation 13, although it still requires the 1-to-1 correspondence, but is E($n$)-invariant. Furthermore, we require our autoencoders to be persistent, namely, it should always be possible to correctly decode a graph $\mathcal{G}$ after a sufficient number of time steps. This is a particular feature of our model, that differ from standard autoencoders, which are not trained to be stable over time. To promote persistency, we use a *multi-target* replay strategy—similar to the one-target version in Section 4.1—so as to ensure an adequate exploration of the state space during training. Specifically, after every optimisation step, we replace the reached state $[\mathbf{X}', \mathbf{H}']$ with the pool state that originated it, and randomly re-initialise pool states after a given number of maximum replacements so as to avoid catastrophic forgetting.

**A 3D demo**  In a first demo experiment, we use COMM-S and PLANAR-S and set $n = 3$ so as to visualise automaton trajectories in 3D. The experiment shows persistent autoencoding, *conditional* generation of 3D point clouds and graph drawing abilities (Eades, 1984; Tamassia, 2013). We randomly sample $t$ in $[15, 25]$

---

[6]Injecting Gaussian noise as initial node features has originally been proposed by Liu et al. (2019), and then also used as a way of overcoming the symmetry problem and over-smoothing (Satorras et al., 2021b; Sato et al., 2021; Godwin et al., 2022).

| | E($n$)-GNCA | EGNN$_4$ | EGNN$_{30}$ | GNCA | GNN$_4$ | GNN$_{30}$ |
|---|---|---|---|---|---|---|
| COMM-S | 1.00±0.00 | 0.91±0.03 | 1.00±0.00 | 0.95±0.01 | 0.88±0.04 | 1.00±0.03 |
| PLANAR-S | 0.99±0.01 | 0.83±0.01 | 0.99±0.02 | 0.88±0.01 | 0.82±0.05 | 0.98±0.03 |
| PLANAR-L | 0.98±0.01 | 0.88±0.35 | 0.99±0.03 | 0.85±0.09 | 0.84±0.25 | 0.97±0.05 |
| PROTEINS | 0.95±0.04 | 0.84±0.01 | 0.97±0.03 | 0.82±0.08 | 0.82±0.03 | 0.95±0.05 |
| SBM | 0.92±0.02 | 0.76±0.01 | 0.96±0.04 | 0.86±0.11 | 0.75±0.07 | 0.93±0.03 |

Table 1: Autoencoding results. F1 scores (the higher the better) averaged over 10 different runs. (E($n$)-) GNCAs are evaluated at time $t = 100$. More details can be found in Figure B.11

at each optimisation step. We reach an average and *persistent* F1 score of 0.98 and 0.96 for COMM-S and PLANAR-S respectively over 10 different runs. Figure 3 shows the learned dynamics of our autoencoder.

**Autoencoding Results** E($n$)-GNCA autoencoders can scale to higher Euclidean spaces and significantly larger graphs, without increasing the size of the models nor losing persistency. We set $n = 8$ for all datasets except SBM where it is set to 24, and randomly sample $t$ in [25, 35]. We compare against standard 4- and 30-layered GNNs (Kipf et al., 2018), 4- and 30-layered EGNNs, and GNCAs (Grattarola et al., 2021). For a fair comparison, we do *not* allow fully connected GNN computational graphs, as opposed to Satorras et al. (2021b), but instead use as computation graph the same graph to autoencode. Table 1 reports autoencoding results. Remarkably, E($n$)-GNCA outperforms a 4-layered (E)GNN and achieves a comparable level of performance of a 30-layer (E)GNN. Several examples of graph reconstructions are available in Figure B.10. Different from standard (E)GNN autoencoders, E($n$)-GNCA autoencoders also exhibit persistent dynamics (cf. Figure B.11) for all datasets except SBM, which, given the variable clustered topology, represents the most challenging dataset.

**E($n$)-GNCAs are multi-target** E($n$)-GNCA autoencoders are multi-target as they can reach many target states, contrary to what is shown in our previous task and previous work (Grattarola et al., 2021; Mordvintsev et al., 2020; 2022). We suppose this to be the consequence of a more relaxed training objective (cf. Equation 15) than the previous one (cf. Equation 13). In graph autoencoding, in fact, target states are *not* explicitly given but rather a *condition* that they must satisfy is (cf. Equations 14 and 15). Therefore, since we are only interested in reconstructing the ground-truth $A$ via Equation 14, E($n$)-GNCAs can converge to any possible configuration from which decoding is possible.

### 4.3 Simulation of E($n$)-equivariant dynamical system

We show the applicability of E($n$)-GNCAs as simulators of E($n$)-equivariant dynamical systems. The goal is to learn the transition rule underlying observed trajectories. Specifically, we train E($n$)-GNCAs to simulate the Boids Algorithm (Reynolds, 1987), a Markovian and distributed multi-agent system designed to simulate flocks of birds using a set of hand-crafted rules. The graph $\mathcal{G}$ is obtained as a fixed-radius nearest neighbourhood of the nodes at each time step—$\mathcal{G}$ changes dynamically over time. This dynamical system (i) can be formulated as a GCA (cf. Equation 1) and (ii) is E($n$)-equivariant. This task is very different from the previous ones as we want to approximate existing dynamics and not discover some that converge to a given state.

**Dataset** We extend the 2D simulation of Grattarola et al. (2021) to a 3D space. We create a dataset of 500 trajectories using the ground-truth simulator. Each trajectory has a duration of 500 time steps and is obtained by evolving 100 boids initialised with random positions and velocities.

**Training** We use attention weights (cf. Equation 8) and Equations 9 and 10 to explicitly account for velocities. We create a mini-batch of randomly sampled sub-trajectories of length $L = 20$. For each mini-batch sub-trajectory $[\mathbf{X}^{(\ell)}, \mathbf{V}^{(\ell)}]_{\ell=1}^{L}$ we input $\tau_\theta$ with state $\mathbf{S}^{(1)} = [\mathbf{X}^{(1)}, \mathbf{V}^{(1)}, \mathbf{H}^{(1)}]$ and run it for $L - 1$ steps, obtaining predicted states $[\mathbf{X}'^{(\ell)}, \mathbf{V}'^{(\ell)}, \mathbf{H}'^{(\ell)}]_{\ell=2}^{L}$. Finally, we optimize the MSE of the estimated velocities with the ground truth ones, i.e.

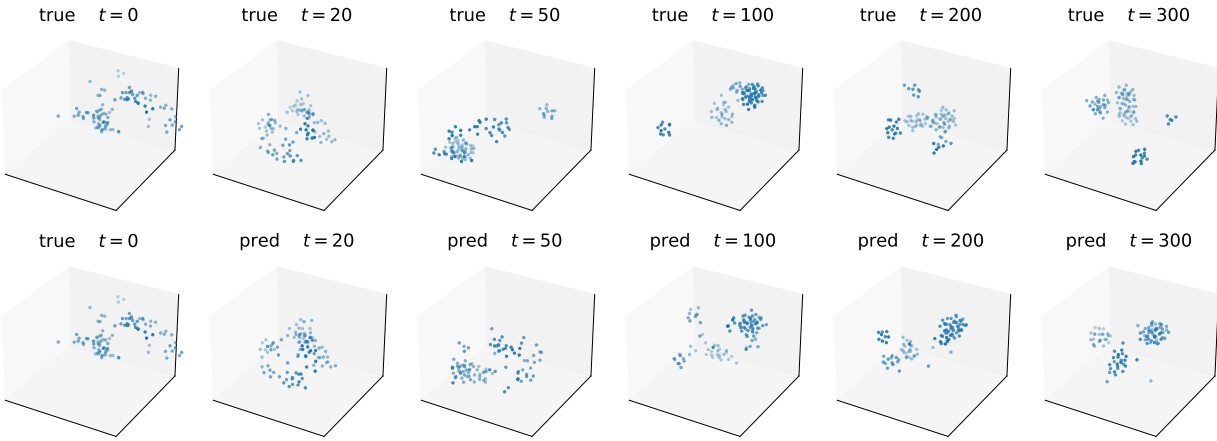

Figure 4: Boids simulation. First (Second) row shows a ground-truth (predicted) trajectory at different time steps. E($n$)-GNCA learns a flocking behaviour similar to the target system, although with smoother and less precise trajectories.

$$\mathcal{L}_{\mathsf{MSE}} = \sum_{\ell=2}^{L} \|\mathbf{V}^{(\ell)} - \mathbf{V}'^{(\ell)}\|^2. \tag{16}$$

As Satorras et al. (2021b), node features $\mathbf{H}^{(1)}$ are initialised as the output of a linear layer taking $\|\mathbf{V}^{(1)}\|$ as input.

**Results**   As already pointed out by Grattarola et al. (2021), one key aspect of simulating continuous (and chaotic) dynamical systems with GNCAs is that small errors in prediction will quickly accumulate, making it almost impossible for the model to perfectly simulate the true dynamics. Therefore, despite reaching a small validation error, the model *cannot* perfectly approximate the true trajectories. However, following Grattarola et al. (2021), we can quantitatively evaluate the quality of the learned transition rule by using the sample entropy (SE) (Richman & Moorman, 2000) and correlation dimension (CD) (Grassberger & Procaccia, 1983), two measures of complexity for real-valued time series. On average, ground-truth trajectories (of length 500) report an average SE and CD of $0.04 \pm 0.01$ and $1.02 \pm 0.22$ respectively, whereas E($n$)-GNCA trajectories report $0.04 \pm 0.02$ and $1.08 \pm 0.15$ for the same measures. The closeness of the measures indicates that E($n$)-GNCA trajectories generate an amount of information comparable to the ground-truth ones, therefore capturing the essence of the underlying rule, Figure 4.

Note that original GNCAs also succeed at capturing the Boids rule. However, them being aware of the global locations, the task is easier to solve. Crucially, moreover, the GNCA model would be unable to work in different reference frames except for the one used during training (e.g. $[0,1]^3$), while our model can transfer, by design, to new frames.

## 5   Discussion

We introduced E($n$)-GNCAs, isotropic automata showing and promising a wide range of applicability. E($n$)-GNCA local interactions have been proven powerful enough to reach globally consistent target conditions and capture complex dynamics. To the best of our knowledge, this is the first work proposing isotropic-by-design graph neural cellular automata.

**Scope of the paper**   This work does *not* focus on (E)GNNs, nor physical simulation, nor graph autoencoding. Instead, our focus is on designing and learning CA rules, as well as showcasing the properties of local, isotropic,

open-ended and distributed self-organizing systems. To give context, note that in NCA papers (Mordvintsev et al., 2020; Palm et al., 2022), the focus is not on image generation but image generation *through emergent computation*. All our experiments are designed to answer the crucial question underlying CAs: How can we design a CA transition rule that behaves according to some high-level specification (e.g. forming a grid)? Since in general the answer to this question requires some (impossible) complex engineering, recent literature leveraged neural networks to learn these rules.

**Isn't it just an EGNN layer?** We make no claim of novelty in the design of EGNN itself (except Equation 9), but rather in the way that EGNN can be trained and used to implement the open-ended local computation that characterizes CAs. However, our model differs from EGNNs in very fundamental aspects like the training procedure, the open-ended inference, and the uncommon single-layered architecture. Moreover, by definition, NCAs are CAs in which the transition rule is parametrized by a neural net: In original NCAs (Mordvintsev et al., 2020; Palm et al., 2022) (resp. GNCAs (Grattarola et al., 2021)), the neural net is a composition of convolutional layers (resp. message-passing layers). In our paper, the neural net consists of an EGNN layer. The chosen neural net endows the NCA with inductive biases that allow it to model specific transition rules and self-organizing systems. The study of these families lies at the core of the NCA literature.

**Broader scope** The possible implications of our work are evident when considering that distributed and self-organizing systems are ubiquitous both in nature and technology (Collinet & Lecuit, 2021). Furthermore, isometries are very common in dynamical systems (e.g. swarming (Reynolds, 1987), particle simulations (Kipf et al., 2018)), active matter (Cichos et al., 2020), and in many practical applications (e.g. point cloud processing, 3D molecular structures (Ramakrishnan et al., 2014)). Our model and its inductive biases are particularly useful in all these scenarios, since they allow to learn and discover—rather than hand-design—the transition rules underlying these systems, while accounting for symmetries.

Notably, one of the most remarkable demonstrations of self-organisation can be found in swarm robotics and active matter modeling (Brambilla et al., 2013; Vicsek et al., 1995). Nowadays, we can program tiny robots to locally interact and form a given pattern, as Mergeable Nervous Systems (Mathews et al., 2017) and Kilobots (Rubenstein et al., 2012) demonstrated. To the best of our knowledge, such programs are currently designed by humans. Our work can enable a line of research in which GNNs further unlock the power of GCAs to implement a desired behavior through differentiable, distributed and emergent computation.

**Limitations & Future Work** Training E($n$)-GNCAs is not easy: We faced problems such as exploding gradients, which we mitigated using weight decay and gradient clipping. Furthermore, the 1-to-1 correspondence between initial and target nodes does **not** make Equations 13 and 15 node-permutation invariant, and this represents a sub-optimal design choice as it leads to abrupt dynamics especially in the first time steps. In future work, we aim to drop this correspondence by adopting ideas from Optimal-Transport (Peyré & Cuturi, 2019; Alvarez-Melis et al., 2019). Furthermore, we plan to use more expressive GNNs (Joshi et al., 2023; Thomas et al., 2018; Batatia et al., 2022; Fuchs et al., 2020), and scale to even bigger graphs using structured seeds (Mordvintsev et al., 2022).

### Acknowledgments

The Eindhoven University of Technology authors received support from their Department of Mathematics and Computer Science and the Eindhoven Artificial Intelligence Systems Institute.

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

# A   Pattern Formation

## A.1   Implementations details

**Model**   Let $n, h$ and $m$ respectively be coordinate, hidden state and message dimensionality. We set $h = 16$, $m = 32$, and $n = 2$ or $n = 3$ depending on the target geometric graph. We normalise node features **H** using PairNorm (Zhao & Akoglu, 2020). Our MLPs (5,300 parameters overall) are defined as follows:

- Message MLP $\phi_m : \mathbb{R}^{2h+1} \rightarrow \mathbb{R}^m$ (cf. Equation 3):

$$\big[\mathsf{LinearLayer}(2h + 1, m), \mathsf{TanH}(), \mathsf{LinearLayer}(m, m), \mathsf{TanH}()\big],$$

- Coordinate MLP $\phi_x : \mathbb{R}^m \rightarrow \mathbb{R}^1$ (cf. Equation 5):

$$\big[\mathsf{LinearLayer}(m, m), \mathsf{TanH}(), \mathsf{LinearLayer}(m, 1), \mathsf{TanH}()\big],$$

- Hidden state MLP $\phi_h : \mathbb{R}^{h+m} \rightarrow \mathbb{R}^h$ (cf. Equation 7):

$$\big[\mathsf{LinearLayer}(m + h, h), \mathsf{TanH}(), \mathsf{LinearLayer}(h, h)\big].$$

**Training**   We train the model by minimising Equation 13, using Adam (Kingma & Ba, 2015) with initial learning rate of 0.0005. The learning rate is then decreased during training using a reduce-on-plateau schedule. We use gradient clipping and weight decay as regularisation techniques. We increase the batch size during training (from a minimum of 4 to 32) so as to promote a faster convergence, and that is because a small batch size leads to more frequent re-initialisation of the pool states in the early training steps. We train the model to convergence, by monitoring the validation loss with a fixed patience of training steps. For all details, we refer the reader to our code. All experiments are run on an NVIDIA Quadro P1000 16GB.

## A.2   Full trajectories & Visualizations

In this subsection, we report the full trajectories of our model for the following geometric graphs: a Line Figure A.1, a 2D grid Figure A.2, an X-shaped pattern Figure A.3, a 3D torus Figure A.4, a 3D cube Figure A.5, a 3D pyramid Figure A.6, a 3D vase Figure A.7 and the Stanford bunny Figure A.8. Furthermore, a visual example of the key feature of anisotropic pattern formation is showed in Figure A.9.

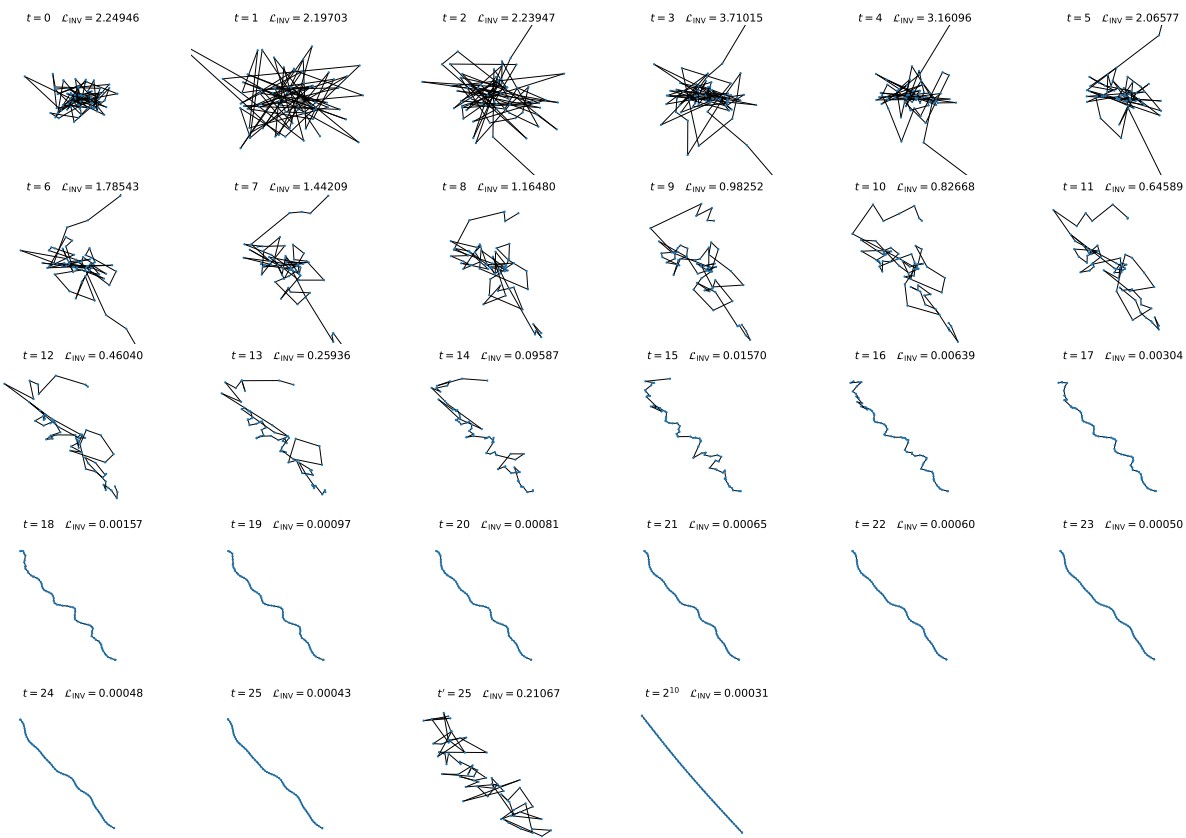

Figure A.1: Convergence to a Line. We report the loss value (Equation 13) in each figure. Global damage occurs at $t' = 25$. Best viewed digitally and zoomed in.

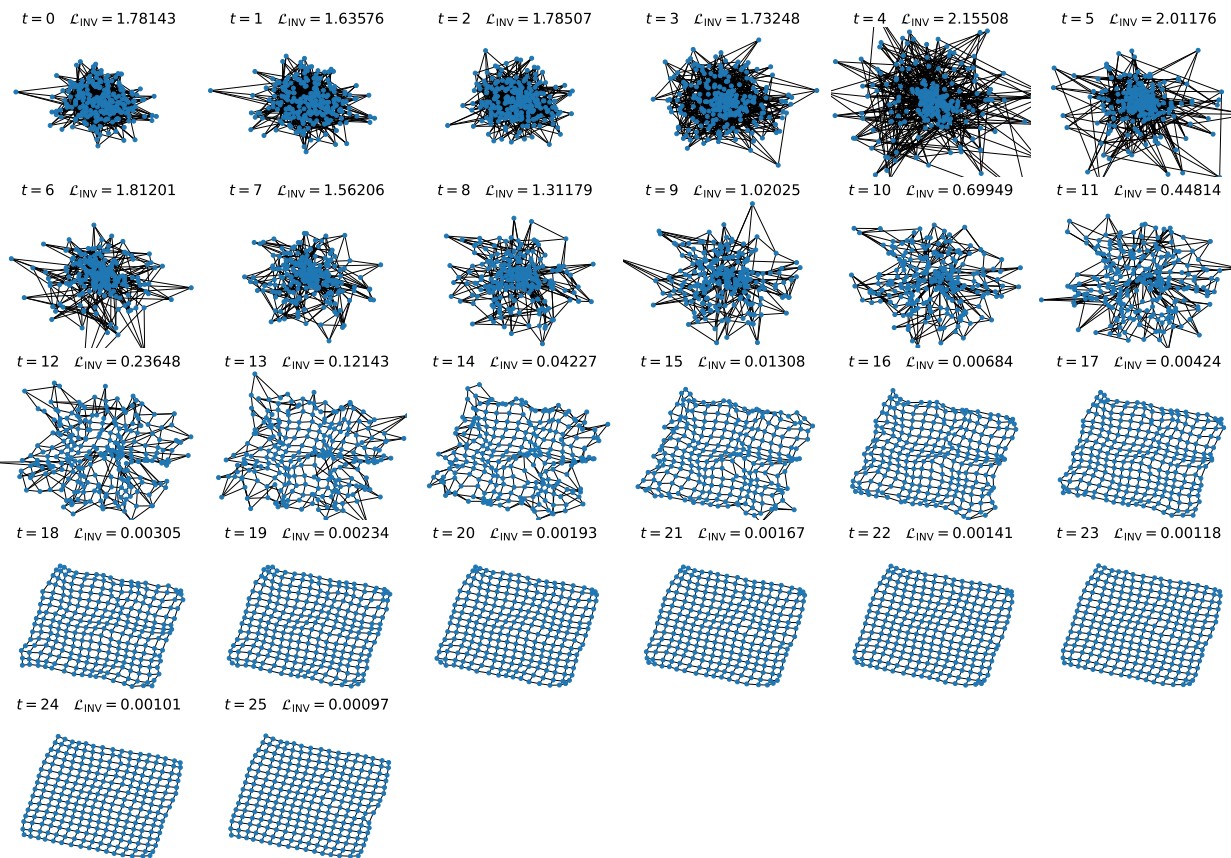

Figure A.2: Convergence to a 2D grid. We report the loss value (Equation 13) in each figure. Best viewed digitally and zoomed in. Regeneration and persistency are showed in Figure 2

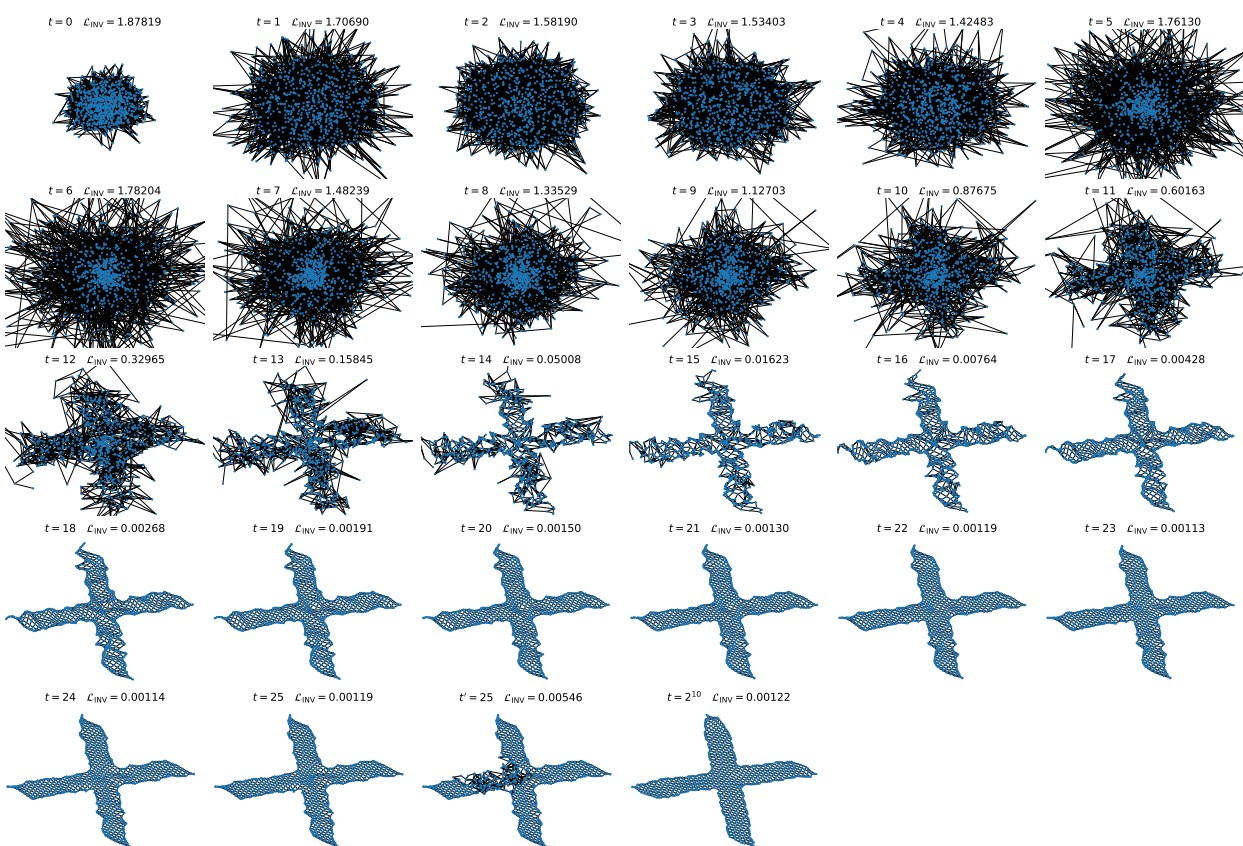

Figure A.3: Convergence to an X-shaped pattern. We report the loss value (Equation 13) in each figure. Damage occurs at $t' = 25$. Best viewed digitally and zoomed in.

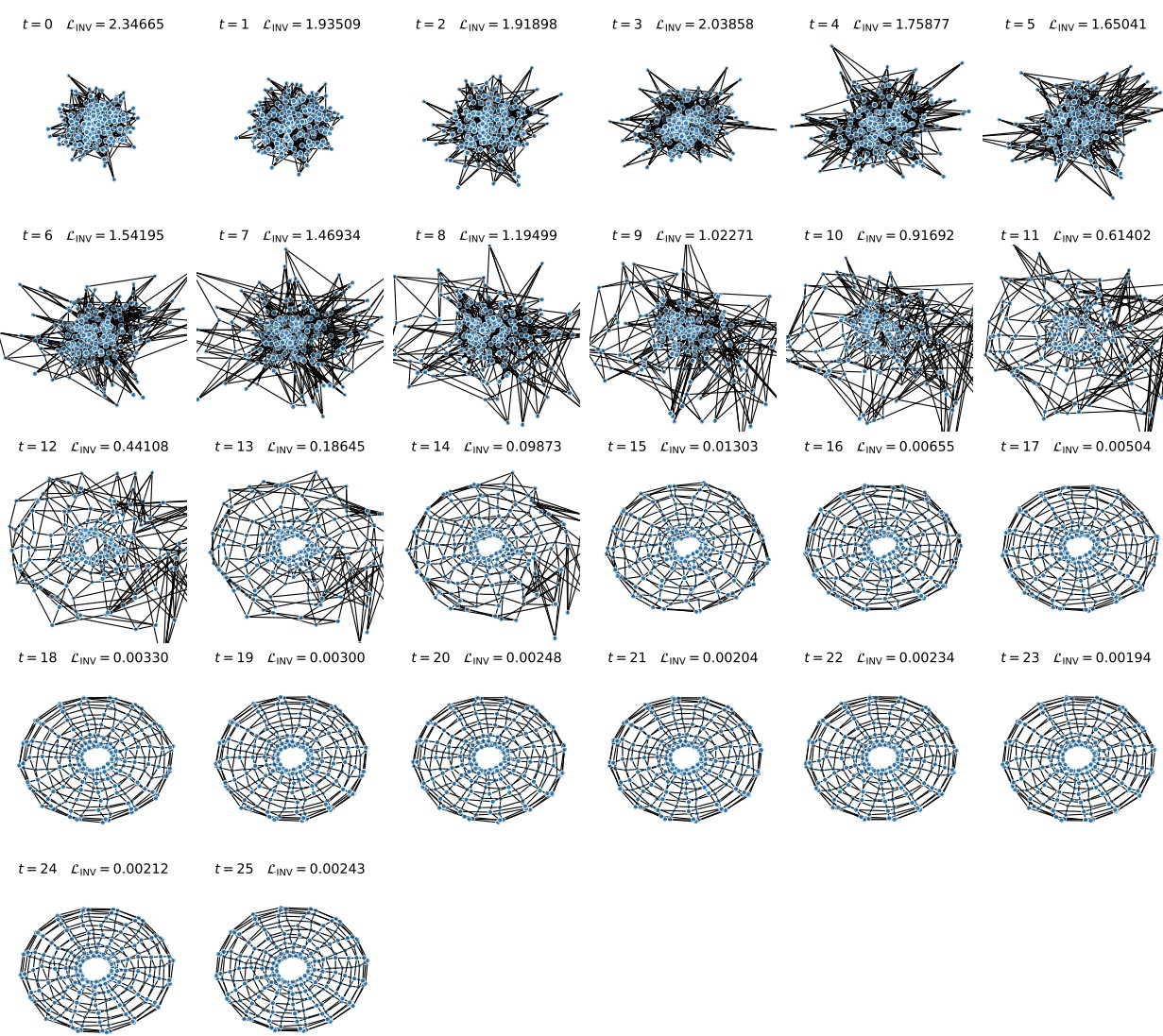

Figure A.4: Convergence to a 3D torus. We report the loss value (Equation 13) in each figure. Best viewed digitally and zoomed in. Regeneration and persistency are showed in Figure 2

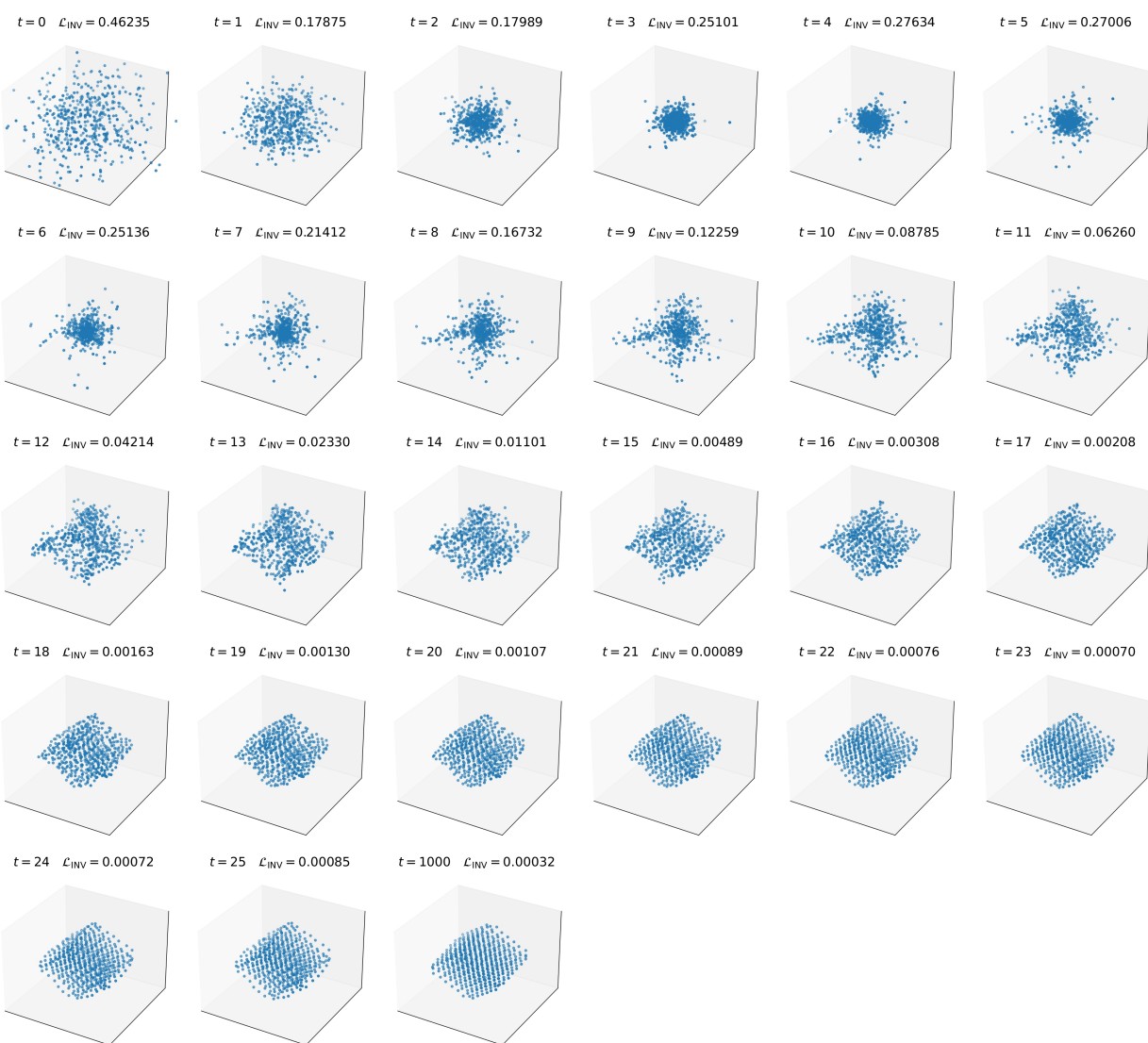

Figure A.5: Convergence to a 3D Cube. We report the loss value (Equation 13) in each figure. To avoid clutter, nearest-neighbor edges are not shown. Best viewed digitally and zoomed in.

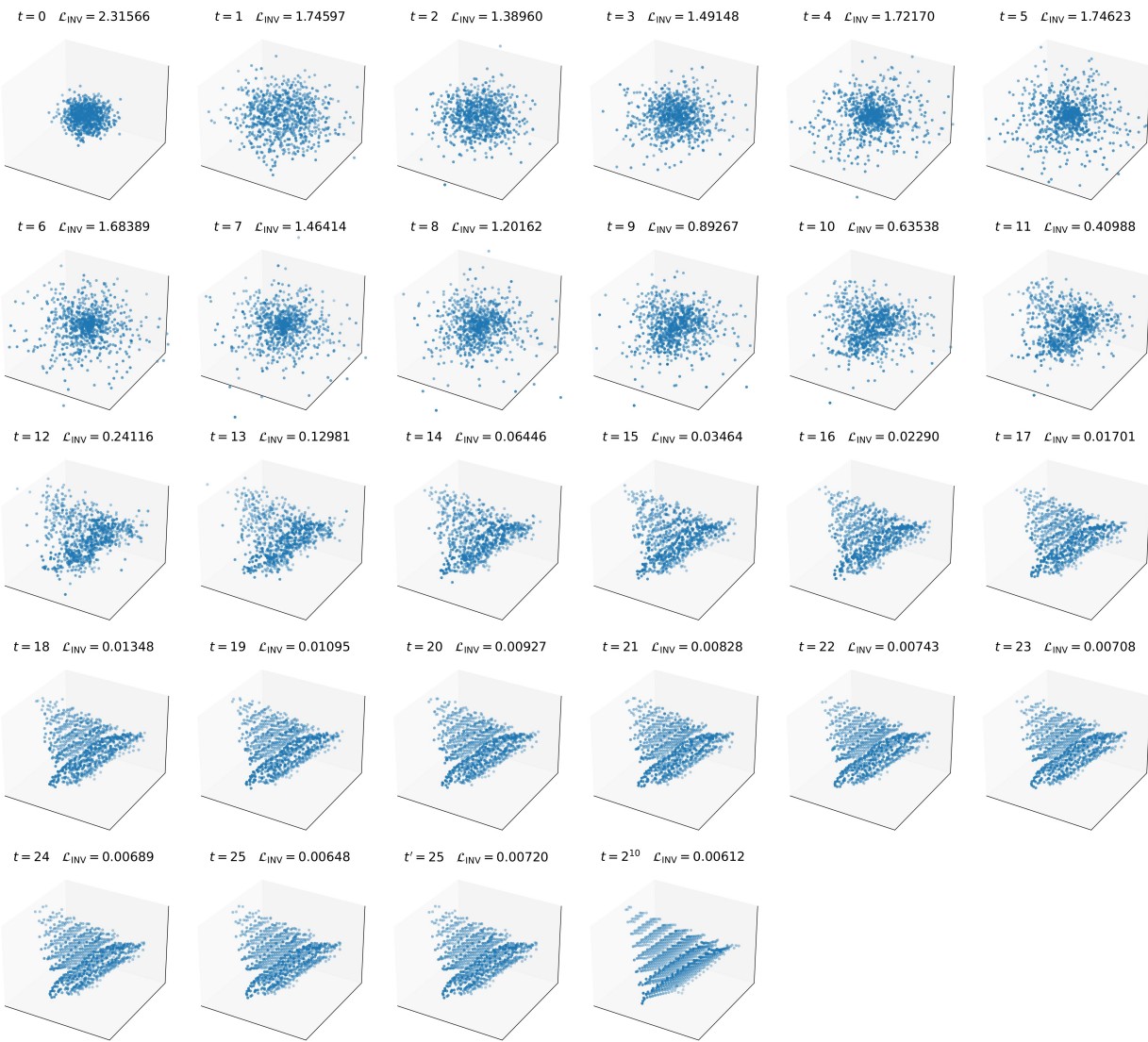

Figure A.6: Convergence to a 3D pyramid. We report the loss value (Equation 13) in each figure. Damage occurs at $t' = 25$. To avoid clutter, nearest-neighbor edges are not shown. Best viewed digitally and zoomed in.

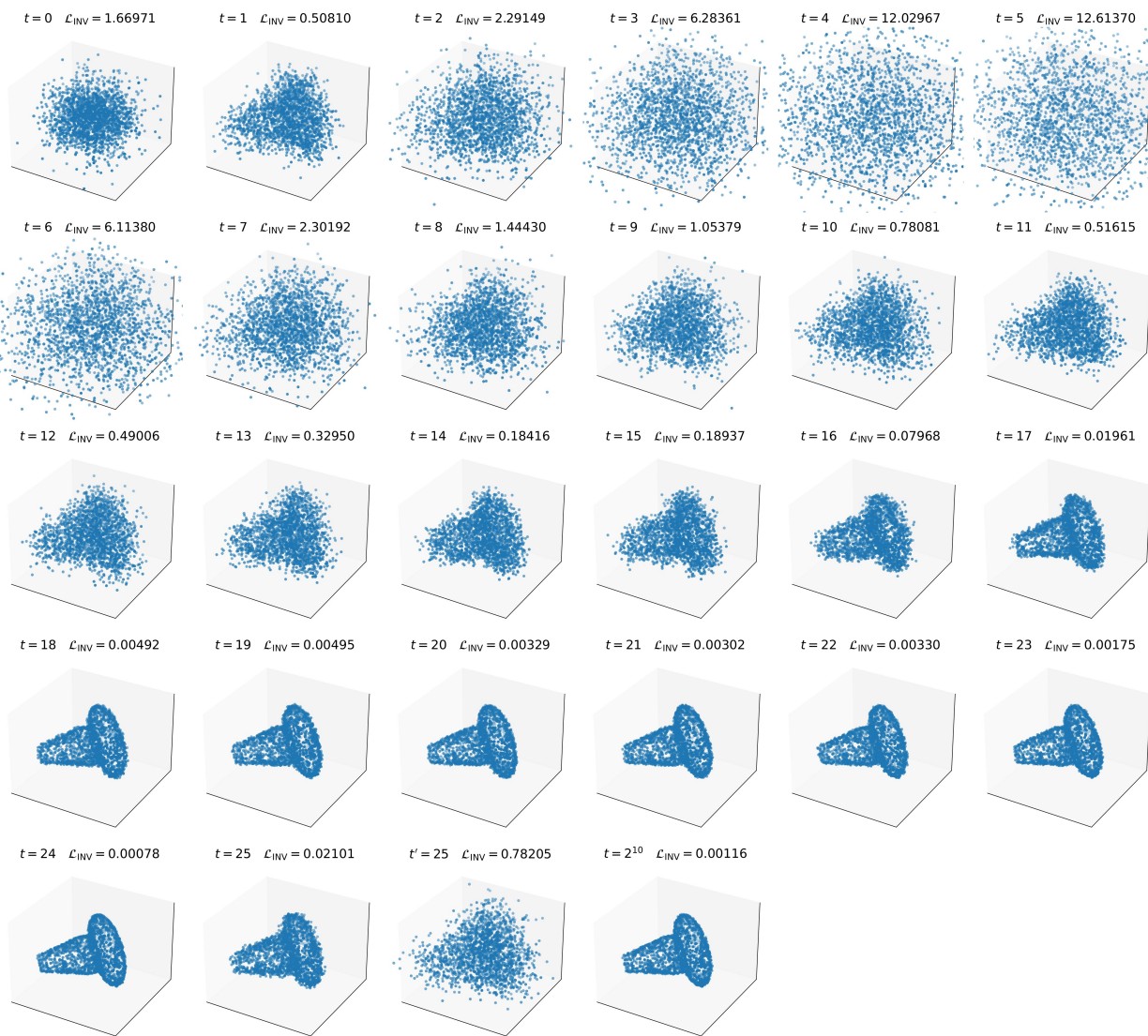

Figure A.7: Convergence to a 3D vase from Shapenet (Chang et al., 2015). We report the loss value (Equation 13) in each figure. Global damage occurs at $t' = 25$. To avoid clutter, nearest-neighbor edges are not shown. Best viewed digitally and zoomed in.

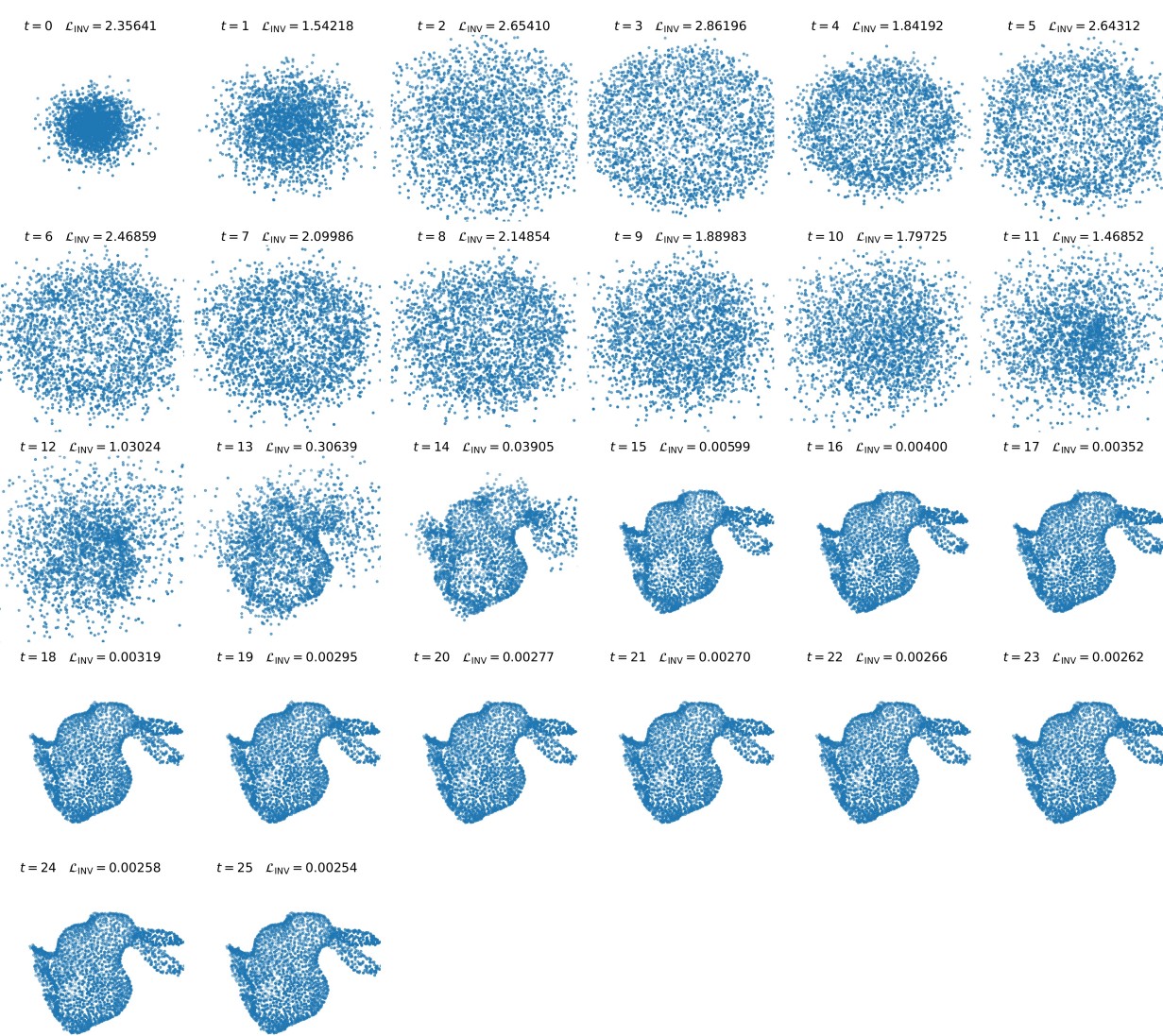

Figure A.8: Convergence to the Stanford bunny. We report the loss value (Equation 13) in each figure. To avoid clutter, nearest-neighbor edges are not shown. Best viewed digitally and zoomed in. Regeneration and persistency are showed in Figure 2

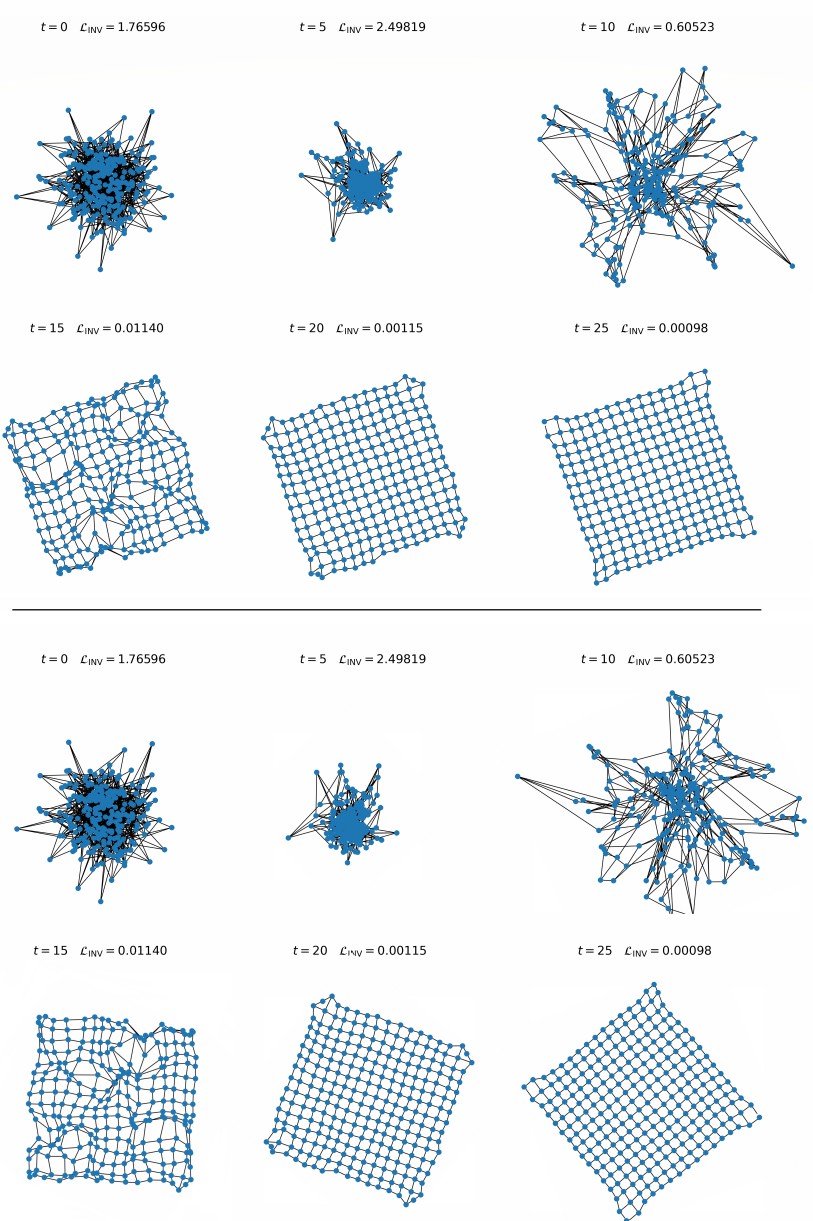

Figure A.9: **Anisotropic grid pattern formation.** First two rows show the formation of a 2D grid starting from a random initial state at different time steps. Last two rows show the same except that before applying the model transition rule $\tau_\theta$ to go from $\mathbf{S}_t$ to state $\mathbf{S}_{t+1}$, we apply a random isometry to all the nodes. The image aims to show that there is no frame dependency and no awareness of global locations: The model acts as if nothing had happened since nodes are only aware of their relative positions to their neighbors. This behaviour is still valid both when perturbations of the nodes occur and when more complex patterns are used (e.g. Stanford bunny), but easier to show with a simple grid. Such behaviour is not possible with original GNCAs (Grattarola et al., 2021).

## B  Graph autoencoding with E($n$)-GNCAs

### B.1  Implementation details

**Model**  We use the exact same architecture as the one detailed in Appendix A.

**Training**  We train the model to minimise Equation 14, using Adam (Kingma & Ba, 2015) with initial learning rate of 0.0005. The learning rate is then decreased during training using a reduce-on-plateau schedule. We set the batch size to 32 and train the model by monitoring the validation loss with a patience of 20 epochs. We use negative edge sampling when computing the loss so as to have a balanced supervisory signal when processing sparse graphs. For all details, we refer the reader to our code.

**Testing**  In order to effectively evaluate the quality of a graph reconstruction, we first need to binarize its soft adjacency matrix $\hat{A} \in [0,1]^{|\mathcal{V}| \times |\mathcal{V}|}$ (cf. Equation 14). Therefore, at test time, we fine-tune a threshold $\hat{t} \in (0,1)$ on the validation set as to maximize the F1 score of validation-set graph reconstructions. Once we have threshold $\hat{t}$, we can binarize soft adjacency matrices of test-set graphs and then compute the F1 scores thereof.

### B.2  Reconstructions

Figure B.10 shows examples of test-set reconstructions from our autoencoding task (cf. sections 4.2).

### B.3  Persistency test

Figure B.11 shows the F1-score trend over time for E($n$)-GNCA autoencoders (cf. sections 4.2).

## C  Simulation of E($n$)-equivariant system

### C.1  Implementation Details

**Model**  Let $n, h$ and $m$ respectively be coordinate, hidden state and message dimensionality. We set $h = 16$, $m = 32$, and $n = 3$. Our MLPs (5500 parameters overall) are then defined as follows:

- Message MLP $\phi_m : \mathbb{R}^{2h+1} \to \mathbb{R}^m$ (cf. Equation 3):

$$\big[\mathsf{LinearLayer}(2h+1, m), \mathsf{TanH}(), \mathsf{LinearLayer}(m, m), \mathsf{TanH}()\big],$$

- Attention MLP $\phi_a : \mathbb{R}^m \to [0,1]^1$ (cf. Equation 8):

$$\big[\mathsf{LinearLayer}(m, 1), \mathsf{Sigmoid}()\big],$$

- Velocity MLP $\phi_v : \mathbb{R}^{h+1} \to \mathbb{R}^1$ (cf. Equation 9):

$$\big[\mathsf{LinearLayer}(m, m), \mathsf{TanH}(), \mathsf{LinearLayer}(m, 1), \mathsf{TanH}()\big],$$

- Coordinate MLP $\phi_x : \mathbb{R}^m \to \mathbb{R}^1$ (cf. Equation 9):

$$\big[\mathsf{LinearLayer}(h+1, h/2), \mathsf{TanH}(), \mathsf{LinearLayer}(h/2, 1)\big],$$

- Hidden state MLP $\phi_h : \mathbb{R}^{h+m} \to \mathbb{R}^h$ (cf. Equation 7):

$$\big[\mathsf{LinearLayer}(m+h, h), \mathsf{TanH}(), \mathsf{LinearLayer}(h, h)\big].$$

**Training**  We train the model to minimise the MSE between ground-truth and predicted trajectories, using Adam (Kingma & Ba, 2015) with initial learning rate of 0.001. The learning rate is then decreased during training using a reduce-on-plateau schedule. We set the batch size to 16 and train the model by monitoring the validation loss with a patience of 20 epochs. For all details, we refer the reader to our code.

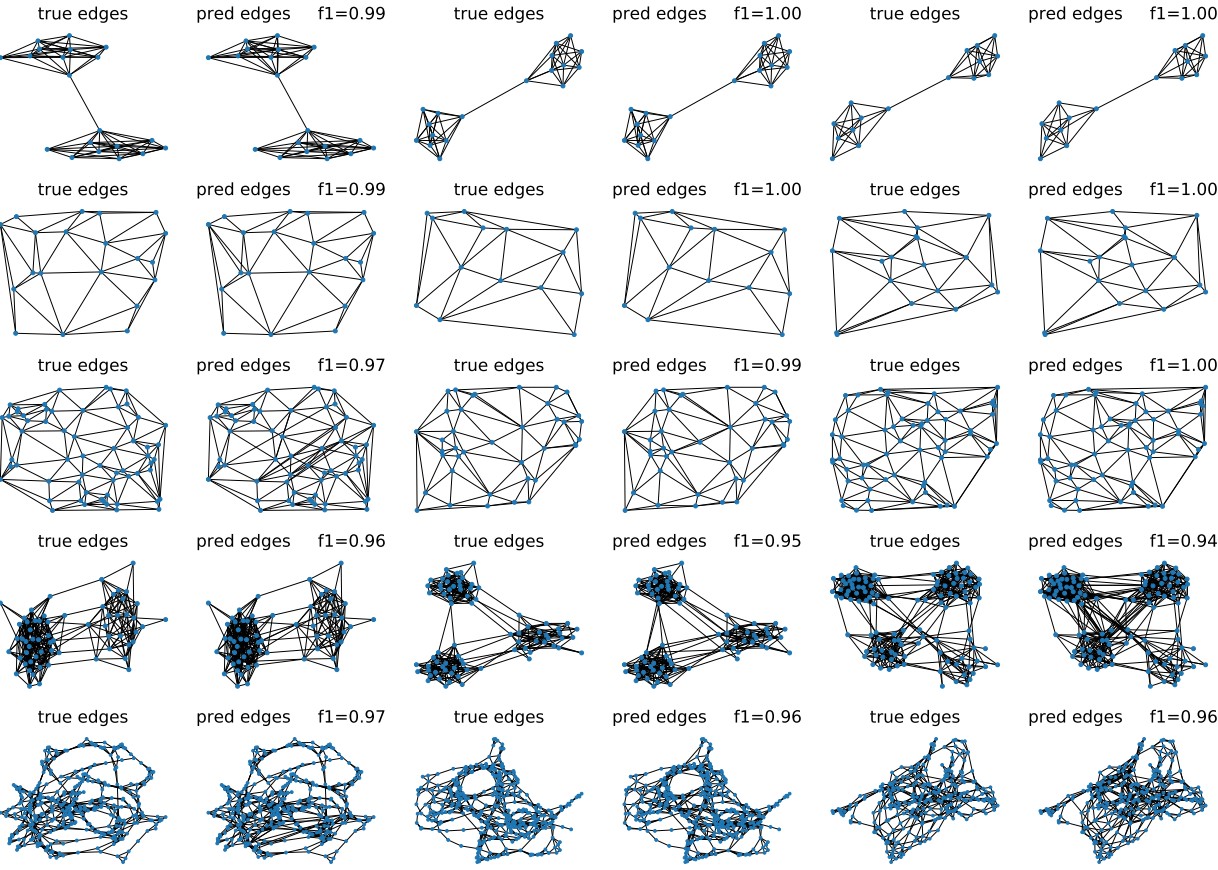

Figure B.10: Test-set graph reconstructions at time step $t = 100$ for COMM-S (1st row), PLANAR-S (2nd row), PLANAR-L (3rd row), PROTEINS (4th row) and SBM (5th row). In alternating columns, we first have ground truth graphs and then respective reconstructions with F1 scores attached on top. The position of the nodes is fixed for both ground-truth and reconstructed graphs so as to visually inspect the quality of the reconstructions. Best viewed digitally and zoomed in.

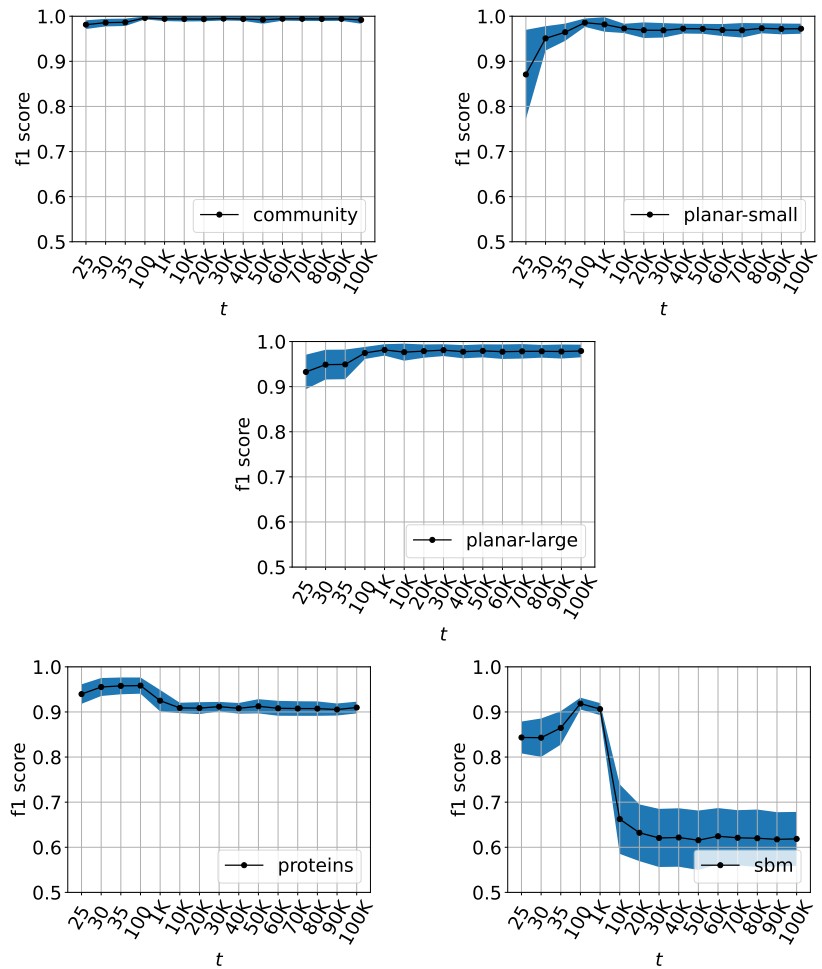

Figure B.11: Persistency test for E($n$)-GNCA autoencoders. We run transition rule $\tau_\theta$ for up to 100,000 time steps and report the F1-score (y-axis) at different time steps (x-axis) for each dataset. Trends are averaged over 10 different runs. E($n$)-GNCAs exhibit a persistent autoencoding trend for all datasets except SBM, which, given its clustered topology, represents the most challenging dataset.

### C.2 The Boids Algorithm

Our implementation of the Boids algorithm is mainly inspired by the one available in AgentPy (Foramitti, 2021), which in turn is based on the seminal work of Reynolds (1987). Each boid has location $\mathbf{x} \in \mathbb{R}^3$ and velocity $\mathbf{v} \in \mathbb{R}^3$. The simulation takes place in a squared 3D fix-sized box. We compute the underlying graph $\mathcal{G}$ at each step of the algorithm by connecting the boids that are within a given radius from each other. At each step of the algorithm, we sequentially apply the following transformations synchronously to all boids to compute the change in each boid velocity and location:

- A *cohesion* force is applied to bring the position of each boid closer to its neighbours;

- An *alignment* force is applied to match the velocity of each boid to the average velocity of its neighbours;

- If the distance between a boid and its neighbours is lower than a user-defined threshold, a *separation* force is applied to steer the boid away from these excessively close neighbours;

- If a boid is within a user-defined radius from the bounding box, a force is applied to steer the boid towards the centre;

- For each boid, the resulting force is summed to the current velocity thus determining the new velocity;

- Finally, the position of each boid is updated according to the new velocity.

We show trajectories in Figure 4. For all details, we refer the reader to our code.

