# OpenReview forum: "E(n)-equivariant Graph Neural Cellular Automata"
_TMLR — Accepted by TMLR_

### Review · Reviewer_fNG5 · 2024-01-25

**Summary Of Contributions:**

The paper addresses the learning of local Cellular Automaton (CA) rules by Neural Nets (Neural CA, NCA) by combining recent ideas:
- Graph NCA (as opposed to NCA that operate on grids)
- E(n)-equivariant GNNs, and in particular the EGCN layer, that are E(n)-equivariant operations, that produce  E(n)-equivariant output.
- A simple graph-attention mechanism (preserving equivariance) is used for an experiment.

Roughly half of the paper is used to introduce these recent ideas, and the other half to present results.

**Audience:**

Yes

**Broader Impact Concerns:**

None, we are far from a recipe to design a killer-bots swarm. Rather, by observing one, this algorithm may help figure out its functioning.

**Claims And Evidence:**

Yes

**Requested Changes:**

## Critical adjustments:
- after Eq (12): when using $E(n)$ and not $SE(n)$, i.e. when including reflections, it is a bit confusing to call $\psi$ "*rigid transformation*". Usually, the term "*rigid motion*" refers to elements of $SE(n)$ (also denoted $E^+(n)$). You could use only the term isometry, which refers to both direct and indirect isometries (chirality + or -). Note: "*rigid transformation*" appears again in page 7.
- "*these properties are a consequence of only processing relative distances and never being aware of absolute node locations*". Here I would replace "*relative distances*" (Scalars) with "*relative positions*" (vectors of $R^n$). In the current form, there is a mistake. Equation 3 (as in Eq (3) of Satorras) is indeed independent of positions and only depends on relative distances (scalars). However, and this is crucial to obtain an equivariant layer (which is strictly more expressive than an invariant one), the relative orientation must be taken into account, as it is the case in Eq (5) (eq (4) of Satorras).
- the graph auto-encoding task (that I am not familiar with) is not presented in a sufficiently pedagogical manner that the paper would be self-sufficient. In the current form it seems the reader has to go read other paper(s) to understand what are these experiments (section 4.2) about.
- section 4.2, "*For a fair comparison, we do not allow underlying fully connected graphs, as opposed to Satorras et al. (2021b)*" I do not understand how the comparison is fair. First, using $25<t<35$ is effectively similar to using 35-layered (Reccurrent) NNs. Comparing a RNN and a feed-forward 4-layers numerical network is not obvious, but that's the spirit of your original approach, so that point is fine, although it's not obvious how many layers should be allowed. Maybe the number such that the number of parameters would be equivalent. The search for and discussion about what is a "*fair comparison*" should be extended.
- (Very important) The loss functions (Eq 13, 15) are implicitly fully-connected-like losses, because they involve all pairs of nodes. Although the single-node output of one iteration step is only computed from the values at the nodes and from the (relative) positions of that node's neighbors (and their features), the parameters updates involve a global knowledge. In other words, although the model (the graph-features update) is strictly local, the teaching is made with global supervision. This kind of subtlety should be addressed (maybe not in sec 4.2 but before, e.g. sec 3). At least, providing the losses early on (sec 3?) could help the reader have a better sense of how E(n)-GNCA works in practice.  This is a crucial point since the conclusions state that "*All our experiments are designed to answer the crucial question underlying CAs: How can we design a CA transition rule that behaves according to some high-level specification (e.g. forming a grid)?*"
- As far as I understand, the loss for the 3rd task (by far the most interesting, in my opinion) is indeed truly local. The (sadly, unnumbered) equation towards the end of page 10 seems to only involve node-related costs, that are simply summed over the graph (since we have weight-sharing over nodes, this global aggregation is of course ok). I think the difference with the 2 previous tasks should be stressed, as it is a crucial claim of the paper (that one can learn strictly local rules -- implicitly, through local learning).
- (Very important) Still about non locality of the losses (13) and (15): I also note that such losses break node-permutation invariance. For instance, in task #1, with eq (13), if I start from the perfect Stanford bunny and swap two nodes, the loss is no longer 0 and the network will attempt to reverse the swap. This is not intuitively what the task is supposed to be: rules are not purely local, each player as a pre-defined role. This should be discussed somewhere. This is also true for Eq (15), as far as I understood this task. On the contrary, task #3 respects the node-permutation symmetry, which is excellent. So again, the claim that rules are local is impeded by the non-locality of the losses used (supervision has a global blueprint, and does not occur through local rules only --except for task #3). I note that currently this is somehow alluded to at the end of the paper: "*the 1-to-1 correspon-dence between initial nodes and target nodes is a non-optimal design choice as it leads to abrupt dynamics especially in the first time steps.*", but the issue is presented as a mere computational issue.
- An important point of clarity: the connectivity of the graphs in 4.1, 4.2 are not provided. Are the GNN fully connected networks ? Or is connection based on distance ? But distance changes over the course of iterations.. so which one is used for back-propagation ? In 4.3, it is specified: "*The graph G is obtained as a fixed-radius nearest neighborhood of the nodes at each time step—G changes dynamically over time.*" But in practice, what is the radius used ?


### Non critical but suggested changes:
- it is mentioned quite late in the paper that an attention mechanism will be used for task #3. It think this can be considered as important.
- section 1 and 2 are rather clear, although it was not instantaneous for me to understand what NCA (or GCNA's) are meant to do/how they operate in practice. In particular I was wondering: on what kind of training data do they learn ? I initially though they could learn the local rules (or rather a NN approximation of them) from observing a simulation of a cellular automaton. This is confirmed by your sec 4.3, but is quite different from the experiments 4.1 and 4.2. Probably you could clarify this by presenting a typical use case of NCAs, specifying in particular what is the training data, how training is done if it's not obvious, and how test performance is evaluated (if it is not trivial).
- why specify that NodeNorm and PairNorm are "*parameter-free normalisation techniques*" ? Why is that important for this work ?
- When referring to more evolved rotation-equivariant GNN schemes (you cite Joshi et al., 2023), I think you could refer to Tensor field networks (Thomas et al 2018), to the e3nn library, and maybe also to the recent MACE paper, *MACE: Higher order equivariant message passing neural networks for fast and accurate force fields*). The networks one can build with such layers are much more expressive than the original GCN-layer based networks.
- Table 1: provide the number of parameters for each model ? And maybe also the training time. And also memory footprint. Ideally, CO2 footprint when known.
- the sample entropy and correlation dimensions are not defined. Although they can read from Wikipedia, it is necessary to define them at least in the appendix of the paper, and provide some intuition about what they compute. Currently it is simply stated that they are "*two measures of complexity for real-valued time series*". In practice indeed they seem characterize sets of time series (or clouds of points for CD). I guess the Lyapunov exponent could also be reported.
- About sample entropy and correlation dimension: Having equal or very similar SE and CD is a necessary but not sufficient condition for the time series to be very similar. Although Fig.4 is quite convincing, a more direct comparison, e.g. even the MSE at 1 iteration averaged over a large number of initial configurations, would be a simple measure, where various models could be compared.
- The term "*Broader Impact*" has been used in a different way than intended by TMLR, I suppose. Maybe "Broader Scope" would be better.
- In terms of Broader Scope (I may be naive since I am not familiar with recent works in GCA/CA/NCA), I would have added the potential use to model real systems and in particular active matter, e.g. biological systems (e.g. cells assemblies) that we know respect local rules, but for which we do not know the rules. That is, from observed time series, infer the rules using GNCAs rather than a time-consuming expert hand0cerafting of the rules. In this sense the goal is not to reproduce some designed behavior but rather to have a reduced-order model, that can then be used to probe unseen situations (less costly test than actual experiments).
- the location of fig 4 seems wrong: it appears before table 1, although table 1 refers to sec 4.2 and figure 4 refers to sec 4.3.

**Strengths And Weaknesses:**

Strengths:
- The paper is clearly written. I enjoyed the introduction which presents CAs, GCAs, GNNs and E(n)-GNNs. Though, I am familiar with all these topics (except for GCAs) -- and indeed I found the part on GCA less clear, probably because I didn't know about them beforehand.
- The proposed approach seems to truly address the learning of a local rule, by a local-only (and isotropic) model. This is especially convincingly shown on task 3 (section 4.3).
- The building of GNCA seems at first like a low-activity niche topic. However, I feel that this work has broad concrete applications (as I suggest below, for learning real-life active matter rules)

Weaknesses:
- The losses used (in task 1 and 2) are non-local (involve all pairs of nodes), contradicting some of the claims. See requested changes.
- The losses used (in task 1 and 2) break node-permutation equivariance, contradicting some of the claims. See requested changes.
- Some definitions are not provided, making the paper not self-sufficient, and a few other informations are not provided. See requested changes.

---

> ### Author Response · Authors · 2024-01-31
>
> We thank reviewer for their very detailed and constructive feedback. Below, we mainly address their 'critical adjustments'.
>
> - We will replace “rigid transformation” with “isometry”
> - We will replace “relative distances” with “relative positions”. Many thanks for spotting this subtlety.
> - We will better introduce the task of graph-autoencoding task.
>
> - In the original graph-autoencoding experiments of Satorras et. al., a fully-connected GNN computational graph has been used to decode graphs that were actually sparse (i.e., GNN computation graph and graphs to autoencode were decoupled). In our experiments, different from what commonly done, we wanted to actually have as baseline how good standard 4-layered EGNNs could do using as computation graph the same graph to auto-encode. We will make this clearer in the paper. Finally, although we feel this comparison is not actually central to the point of the paper, we are currently experimenting with a 30-layered EGNNs, and will report results asap.
>
> - Thanks for raising this important point. Please note that at the end of the 2nd paragraph on page 6 we write “we are allowed to express global information within the loss function employed for training”, but we will elaborate more. The “fully-connected” losses are necessary to uniquely characterize the target state, and this cannot be done by only using local information. For instance, consider a simple square: If we only minimize for the distances of its edges, we could end up with a state whose loss value is zero but that does not form a square (e.g., when two opposite vertices share the same location). In other words, there is no other way than accounting for all the pairwise distances to uniquely describe a point cloud. However, this does ***not*** make the model less local: Even if global information is accounted in the loss function, the task can only be solved via local communication.
>
> - *On the difference between tasks 4.1/4.2 and task 4.3* (also related to the 2nd point of "Non critical but suggested changes"): In tasks 4.1 and 4.2 , we do not approximate some ground-truth dynamics but, rather, we discover local transition rules that converge to a particular state of interest. There's no training set, all we have is a geometric graph (e.g. the Bunny). What is being processed at each optimisation step is a batch of previous states to which the automaton converged, where such batch is sampled from a cache/pool that is updated during training. This is very different from how (E)GNNs are commonly trained. For a more interactive explanation of these techniques, we refer to https://distill.pub/2020/growing-ca/. In contrast, in tasks 4.3 we have available some ground-truth trajectories that we want to approximate. We will stress more the difference between these tasks (and we're glad that the reviewer seems to particularly like our task 4.3). We have also added numbering to the equation on page 10.
>
> [to be continued...]

---

> ### Author Response · Authors · 2024-01-31
>
> - We want to sincerely thank the reviewer because this point reflects their reviewing effort. Indeed, Eq. 13 and 15 break node-permutation equivariance, but the model still is *by design*. The crux of the discussion lies in the 1-to-1 correspondence between initial and target nodes. To address this issue, we initially tried to use sliced optimal transport (OT). However, the issue is that such loss despite being node-permutation invariant is ***not*** E(n)-invariant, and therefore would force the automaton to converge to a fixed global orientation of the target state, representing a very bad design choice since the model is unaware of global locations. This is in contrast to eqs. 13 and 15 which despite forcing 1-to-1 correspondence, are E(n)-invariant, and give the model the freedom to converge in any orientation of the target.  We could ***not*** find an E(n)-invariant OT-based loss that is efficiently computable and graph-wise parallelizable. An alternative possible solution could involve *structured seeds*, i.e. using some special fixed-in-space nodes that allow to bid to a specific orientation, therefore justifying application of slided-OT losses without contrasting equivariance (this idea is also used in https://direct.mit.edu/isal/proceedings/isal2022/34/65/112305).
> Summarizing, there are two aspects to disentangle: the architecture and the losses, and we make no claims about the node-equivariance of the latter (but we realize we have to elaborate more). We emphasise that despite the current limitations of the loss functions, we believe that the paper goes a step beyond previous work by solving fundamental GNCA issues. Also, although the losses may not be optimal for learning, it can still be claimed that our rules are truly local because they are by design, not matter the loss functions used. There are definitely better (more natural, well-behaved) rules to be found but yet ours are local.  If the reviewer has more suggestions about how we could address this issue, we would be more than happy to explore this in future work. (Last remark: Note that even if we used OT-based losses, we would account for 'global information': for instance, in sliced-OT, the random projections account for all pairs of nodes.)
>
> - In tasks 4.1 and 4.2 the sparse connectivity of the GNN computational graph is kept fixed and equal to a given geometric graph. In contrast, in task 4.3, we change the underlying graph at every step based on distance thresholding, which is commonly done in neural physics simulation papers (e.g., https://openreview.net/forum?id=CCVsGbhFdj) and does not affect backprop. We use a radius of 2.5 in a 3D box of size $[-20, 20]^3$ (see attached code for further details).
>
> - We finally want to thank the reviewer also for their suggested changes, and specifically for the last point discussing potential relevant applications (does the review have references to provide?). All our changes will be highlighted in blue in the revised version(s) of the manuscript asap.

---

> > ### Comment · Reviewer_fNG5 · 2024-02-06
> > **acknowledgement**
> >
> > I acknowledge the authors' answer, I thank them for their detailed answers, I agree with everything they say, and now eagerly await their resubmission.

---

> > > ### Author Response · Authors · 2024-02-07
> > >
> > > We uploaded the revised manuscript. We thank again the reviewer for their reviewing effort, and for helping us improving our submission.

---

> ### Comment · Reviewer_fNG5 · 2024-02-13
> **Answer to resubmission**
>
> First, I thank the authors for their thorough answer to my critical comments, and for considering a number of my secondary comments.
>
> My main concern was about the phrasing of the claims (about node-permutation invariance and most importantly about loclaity vs global information), that needed to be reduced or rather, better contextualized.
>
> **The current version of the paper fully adresses my concerns, so I am happy to approve the paper.**
>
>
> ## Main comments
> #### Here are some comments, that the authors may want to use for discussions, or not.
>
> 1. About permutation-invariance and footnote 4, i.e. the square that can be folded along the diagonal, preserving all distances but not achieving the proper final state: this example emphasizes that a good purely local loss should account for neighbor's relative positions AND also account for their (oriented) angles. In that way one could uniquely define the target structure (up to a rigid transformation, and maybe not up to a mirror symmetry, but that's just a factor 2).
>
>    I think that tensor field networks, are the way to go to address this issue, since you could write a loss that depends on (local) relative angles using these networks (and still be equivariant).
>
>     I don't think this needs to be discussed in the paper, altough it could be an opening.
>
> 2. About active matter:
> - I was thinking about something like the fig 5 b and c of "Machine learning for active matter" https://doi.org/10.1038/s42256-020-0146-9 - but then you already mention robots swarms.
> - Also, there are works on flocks (or fish school) where, from real data, people try to derive simple CA rules that reproduce the flocking behavior (see the works of Cavagna, a statistical physicist). An alternative possibility would be to learn these rules using the GNCA approach.
> - You may even cite morphogenesis (some of which is a result of self-organization, see e.g. a paper published in *nature reviews molecular cell biology*: "Programmed and self-organized flow of information during morphogenesis").
>
>      I think that outlining these possibilities would provide additional readership to the paper.
>
> 3. For now, the link to the code is missing. Please use zenodo or similar long-term archives rather than gitlab-style services.
>
> 4. It may be pointless to report the details of the training runtime because it's rather small in all cases. If that is so, please report the order of magnitude (for each task), and the hardware used. This is a good habit that should be used in all works (not just the compute-hungry).
>
> ## Minor comments:
>
> - I still find that the explanations for the graph auto-encoding task from section 4.2 lacks clarity. But this is not crucial to the paper. From Fig. 3 I understand that nodes have a spatial location, i.e. the graph is not featureless (the node feature is at least the node's location). As far as I understand, the goal is to recover the adjacency matrix, from the node features/locations ? But in your case, their are some features (the node location), that is update by $\tau_\theta$, so as to learn to converge to the proper locations, which correspond to a good adjacency matrix, as given in Eq (14) ? A couple of lines to explain that could help the reader understand the task from the first read.
> - there is a typo in eq.1 (a closing parenthesis ")" and a } were swapped).

---

> > ### Author Response · Authors · 2024-02-15
> >
> > We thank the reviewer again for their detailed response, and are happy that the current version of the paper fully addresses their concerns. We will consider their final comments in the camera-ready version of the paper.

---

### Review · Reviewer_EyWQ · 2024-01-31

**Summary Of Contributions:**

In this work, the authors combine prior art on designing graph neural cellular automata (GNCAs) with E(n)-equivariance constraints (following the popular work of Satorras et al.). The inclusion of E(n) symmetries into the model allows for designing an isotropic model, and one that respects the fundamental principle of cellular automata -- that all rule applications are local. Further, the paper also presents a series of benchmarks, investigations and insights on how to train such systems in a stable manner.

**Audience:**

Yes

**Claims And Evidence:**

Yes

**Requested Changes:**

As far as TMLR is concerned, I think this paper is an open-and-shut case. It presents research that has a clear audience (the CA community), with convincing results, and has no outstanding issues with clarity.

Hence, my recommendation will be to accept the paper.

That being said, if the authors would want some ideas for improving the work further---or inspiring future work---here are a few suggestions:

* E(n)-GNNs from Satorras et al. offer only one solution for satisfying E(n)-equivariance, and one that is rather ad-hoc. It would be interesting to see how more exhaustive approaches (e.g., the SE(3)-Transformer) would perform as a base model in this setting. _[Note that SE(3)-Transformers have a notion of neighbourhood, so they would not violate locality.]_
* In my opinion, algorithmic execution tasks may be seen as related to CA---especially if the execution operates over a neighbourhood. It might be interesting to see how the proposed architecture performs on some tasks in the CLRS-30 benchmark. Further, recent work on this benchmark, such as G-ForgetNet (Bohde et al., ICLR'24; https://openreview.net/forum?id=Kn7tWhuetn) explicitly call out and leverage the Markovian property of execution tasks, meaning they may be relatable here as well.

**Strengths And Weaknesses:**

The paper was a delight to read -- simple, clear and straight-to-the-point. The related work is clearly exposed, the paper does a good job delimiting its own contributions with respect to it, and the use of E(n) equivariance to model cellular automata is an elegant and simple idea. The experiments are diverse, relevant for the CA community, and feature nice additions (like modelling unattributed graph distributions). The results are presented in a way that is likely to be useful to researchers building up on this work.

In my opinion, the leading weakness of the work is the fact that, conceptually, it is a really straightforward application of E(n)-GNNs onto the CA domain (of course, this does not take into account the complexities of training equivariant GNNs in this domain). Hence, there is no strong core algorithmic merit to the work, and I would treat it more as applied work: an architecture was identified to have the properties necessary for GNCAs to do better, and its direct application was tested. More architectural investigations and variations -- especially those that may be more closely aligned to the CA domain would have strengthened the work's significance.

---

> ### Author Response · Authors · 2024-01-31
>
> We thank the reviewer for (i) their kind words (e.g. 'the paper was a delight to read -- simple, clear and straight-to-the-point'), (ii) suggestions about future work, and (iii) for recommending acceptance.
>
> Just one comment about the architecture.
> The EGNN architecture was expressive enough to succeed in all tasks we considered, and, therefore, given its cheap computation and the scope of the paper, we decided to stick with it.
> However, we agree that investigating more expressive/alternative architectures (like the one suggested) is definitely a potential future direction, that can potentially lead to better-behaved dynamics.

---

### Review · Reviewer_9ZGL · 2024-01-31

**Summary Of Contributions:**

In this paper, the authors introduce Equivariant Graph Neural Cellular Automata, which is an architecture combining EGNN, that is, GNN that respect additional equivariance/invariance on (part of) their node features such as equivariance to isometric transform, and NCA, which are Recurrent neural net designed to learn particular transition rules. They illustrate their models on learning pattern formation from random noise, learning Graph auto-encoders, and learning dynamical systems.

**Audience:**

Yes

**Claims And Evidence:**

Yes

**Requested Changes:**

A point that could be better explained is the potential "scale ambiguity" in the model, if the pairwise distance are against a trainable weight then both could be multiplied by a factor and its inverse. For instance, in the graph autoencoder part, if both x_i and delta_2 are trainable, then they could be rescaled with no change. I believe it may be handled by the various normalizations implemented by the authors to fight oversmoothing and such, but it deserves to be written.

Some concepts that could also be better explained:
- what is the "locality bottleneck"? Why is it important?
- what is exactly the "inductive bias necessary to model hidden states" mentioned by the authors? Why are location-independent features important for these "hidden states"?

Typo:
- undelying (p2)

**Strengths And Weaknesses:**

Strengths:
The paper is very well-written and pedagogical. The experiments are well-designed, and one of the main strength of the paper is undoubtedly the successful but challenging training of such models, with quite a lot of hacks and tricks both in the architecture itself (layer normalization etc) and in the experimental design itself (the way pools and batches are handled, updated, etc.)

Weaknesses:
As acknowledged by the authors, the model has limited novelty: all its components were already known, and the main novelty lies in the way the layers are stacked in a recursive fashion and trained.

---

> ### Author Response · Authors · 2024-01-31
>
> We thank the reviewer for highlighting the strengths of our work, and specifically for promoting the way in which we train our models
> (an aspect that we feel can be easily overlooked but that is actually paramount). We reply below to reviewer's discussion points:
>
> - On *scale ambiguity*: We genuinely did not fully get this point and we assume the reviewer meant "delta_1" instead of "x_i" (since coordinate x_i cannot be trained).
> We believe this point concerns the fact that distances can be re-scaled while still representing the same shape; in other words, shapes are scale invariant.
> Indeed, given the maximum and minimum possible distances, we could min-max normalize all pairwise distances and use appropriate scale coefficients to form a bigger/smaller shape.
> Are we on point or not? We're happy to discuss more.
>
>
> - On *locality bottleneck*: The concept of "locality bottleneck" in the context of cellular automata refers to a strict limitation or constraint on how information propagates through the system: Nodes are influenced only by their immediate surroundings and do not have direct access to the global state of the entire automaton. Although at first glance this *seems* to be true also for standard (E)GNNs, it is actually very common to use fully-connected computational graphs to process sparse graph (as done in the E(n)-GNN paper from Satorras et al.) or to use sparse computation graphs but relying on global graph features/embeddings.
> In contrast, our model, despite possibly receiving a training signal that accounts for global information, can only converge to a global state of interest via local communication. Such constraint basically creates a communication bottleneck since the tasks we consider would be much easier (almost trivial) to solve with fully-connected graphs and/or having access to global information when applying the transition rule.
>
> - On *hidden states*:  Quoting [A]: "Hidden channels don’t have a predefined meaning, and it’s up to the update rule to decide what to use them for. They can be interpreted as concentrations of some chemicals, electric potentials or some other signaling mechanism that are used by cells to orchestrate the growth." In our specific case, location-independent features (aka, hidden states) are crucial because they encode past evolutionary history and other geometric information.
> Also note that in the original GNCA framework, ***global node locations*** represented the whole state of a single node, and the transition rule could update the global position of a node based on its global position and those of its neighbors.
> Therefore, in previous GNCA work, there's no 'hidden' state and no possibility to encode other kind of information. In our case, instead, each node $i$ has a hidden state $\mathbf{h}_i$, and the transition rule can update the position of a node based on its hidden state, those of its neighbors, plus all the relative positions.
>
> [A] Mordvintsev, Alexander, et al. "Growing neural cellular automata." Distill 5.2 (2020): e23.

---

### Decision · Action_Editor_Q153 · 2024-02-25

**Recommendation:** Accept as is

**Comment:**

This paper has been praised by reviewers and clearly deserves to be published at TMLR. Congratulations!

**Audience:**

I anticipate that this paper will be of interest to a fraction of the TMLR broad audience.

**Claims And Evidence:**

All claims in the paper are clearly supported by appropriate evidence, as praised by the reviewers.